# A phase II open label clinical study of the safety, tolerability and efficacy of ILB® for Amyotrophic Lateral Sclerosis

Ann Logan[1,2]*, Zsuzsanna Nagy[3], Nicholas M. Barnes[3], Antonio Belli[3], Valentina Di Pietro[3], Barbara Tavazzi[4,5], Giuseppe Lazzarino[6], Giacomo Lazzarino[7], Lars Bruce[8], Lennart I. Persson[9]

**1** Axolotl Consulting Ltd, Droitwich, United Kingdom, **2** Division of Biomedical Sciences, Warwick Medical School, University of Warwick, Coventry, United Kingdom, **3** College of Medical and Dental Sciences, University of Birmingham, Birmingham, United Kingdom, **4** Department of Basic Biotechnological Sciences, Intensive and Perioperative Clinics, Catholic University of Rome, Rome, Italy, **5** Fondazione Policlinico Universitario A. Gemelli IRCCS, Rome, Italy, **6** Department of Biomedical and Biotechnological Sciences, Division of Medical Biochemistry, University of Catania, Catania, Italy, **7** UniCamillus, Saint Camillus International University of Health Sciences, Rome, Italy, **8** Tikomed AB, Viken, Sweden, **9** Department of Clinical Neuroscience, Institute of Neuroscience and Physiology, The Sahlgrenska Academy, University of Gothenburg, Gothenburg, Sweden

* ann.logan@warwick.ac.uk

**Data Availability Statement:** All relevant data that support the findings in this study are available from the EU Clinical Trials Register (available from: https://www.clinicaltrialsregister.eu/ctr-search/trial/

## Abstract

### Introduction

Amyotrophic lateral sclerosis (ALS) is an invariably lethal progressive disease, causing degeneration of neurons and muscle. No current treatment halts or reverses disease advance. This single arm, open label, clinical trial in patients with ALS investigated the safety and tolerability of a novel modified low molecular weight dextran sulphate (LMW-DS, named ILB®) previously proven safe for use in healthy volunteers and shown to exert potent neurotrophic effects in pre-clinical studies. Secondary endpoints relate to efficacy and exploratory biomarkers.

### Methods

Thirteen patients with ALS were treated with 5 weekly subcutaneous injections of ILB®. Safety and efficacy outcome measures were recorded weekly during treatment and at regular intervals for a further 70 days. Functional and laboratory biomarkers were assessed before, during and after treatment.

### Results

No deaths, serious adverse events or participant withdrawals occurred during or after ILB® treatment and no significant drug-related changes in blood safety markers were evident, demonstrating safety and tolerability of the drug in this cohort of patients with ALS. The PK of ILB® in patients with ALS was similar to that seen in healthy controls. The ILB® injection elicited a transient elevation of plasma Hepatocyte Growth Factor, a neurotrophic and myogenic growth factor. Following the ILB® injections patients reported increased vitality,

2017-005065-47/results) and from the corresponding author on request.

**Funding:** This study at the Sahlgrenska Hospital, Gothenburg, was funded by Tikomed AB (https://www.tikomed.com/) via the Clinical Trials Centre (https://www.gothiaforum.com/ctc). Tikomed AB and/or from Axolotl Consulting Ltd. provided support in the form of consultancy payments for LB, AL, NMB and ZN. The specific roles of these authors are articulated in the 'author contributions' section. Tikomed AB gave permission for publication and reviewed the manuscript prior to publication, but the funder had no additional role in the study design, data collection, and analysis.

**Competing interests:** The authors have read the journal's policy and have the following competing interests: LB and AL are paid consultants of Tikomed AB. The authors would like to declare the following patents/patent applications associated with this research: Patents pertaining to this LMW-DS drug have been filed by Tikomed AB (publication number: WO 2016/076780 – New dextran sulphate) with LB as a co-inventor ILB®. LB is a board member of Tikomed AB and receives consultancy payments for work related to the commercial development of ILB®. AL is affiliated with Axolotl Consulting Ltd. This does not alter our adherence to PLOS ONE policies on sharing data and materials.

decreased spasticity and increased mobility. The ALSFRS-R rating improved from $36.31 \pm 6.66$ to $38.77 \pm 6.44$ and the Norris rating also improved from $70.61 \pm 13.91$ to $77.85 \pm 14.24$ by Day 36. The improvement of functions was associated with a decrease in muscle atrophy biomarkers. These therapeutic benefits decreased 3–4 weeks after the last dosage.

## Conclusions

This pilot clinical study demonstrates safety and tolerability of ILB® in patients with ALS. The exploratory biomarker and functional measures must be cautiously interpreted but suggest clinical benefit and have a bearing on the mechanism of action of ILB®. The results support the drug's potential as the first disease modifying treatment for patients with ALS.

## Trial registration

EudraCT 2017-005065-47.

## Introduction

ALS is a degenerative disease affecting both upper and lower motor neurons in brain, brain stem and spinal cord. The major clinical signs of ALS are weakness and atrophy of voluntary muscles, increased muscular tone with increasing spasticity or flaccid paresis, decreased fine motor skills, as well as increasing difficulties of swallowing, speech and respiration. The incidence varies between 1–3 per 100,000 persons, giving rise to a prevalence of between 4–12 per 100,000 persons [1]. About 10% of affected people have familial ALS with genetic causes, while 90% have sporadic ALS. Genetics and variable risk factors partly influence the rate of progression of ALS [2]. For most subjects, the disease is relentlessly progressive, and death usually occurs because of respiratory insufficiency or pneumonia within 2–5 years of diagnosis [3]. No effective cure exists, and currently approved drugs have only marginal effects, even in the most affected persons.

Neuronal degeneration occurs in ALS, with an increase of neurofilaments (NfL) and other remnants of degenerated neurons measurable in cerebrospinal fluid (CSF) [4, 5]. The degeneration and inflammation seen in the central nervous system (CNS) is reflected by an increase of immunoglobulins, complement and other inflammatory biomarkers in serum, plasma and CSF [6–11], as well as reactive cellular changes in astrocytes and microglia. Besides neuroinflammation, glutamate excitotoxicity and oxidative stress are thought to contribute to the neural cell dysfunction and death [12–14]. As in other degenerative processes, there is activation of the ubiquitine-proteasome system, used for degrading many of the cellular remains of proteins or peptides that are released from damaged CNS and muscle cells into tissue fluids [15–21]. Accordingly, CSF, serum and plasma biomarkers can be used to monitor the degenerative and inflammatory progress of the disease and the possible effects of drug treatment [22].

The low molecular weight dextran sulphate (LMW-DS, ILB®) used in this study (Tikomed AB, Viken, Sweden) is a novel patented formulation of a modified glycosaminoglycan that exerts specific neurotrophic and myogenic effects. Although a limited number of animal studies in neurological disease models have been carried out to date with ILB®, its pre-clinical safety profile is well established and the drug has been shown to restore brain energy metabolism in the injured brain after severe traumatic brain injury in rats [23, 24]. Subcutaneously injected ILB® induces a rapid release of pharmacologically relevant levels of Hepatocyte

Growth Factor (HGF) into the circulation in animals and healthy human volunteers [24], which may provide a key neurotrophic stimulus to the disease-compromised CNS as well as a myogenic stimulus to degenerating muscle [25–29]. Of direct relevance, clinical trials of intrathecal HGF for the treatment of ALS are currently in progress [30, 31]. We hypothesise that ILB® can be safely administered to patients with ALS where it will be well tolerated and might be effective at treating the consequences of progressive neurodegeneration and muscular atrophy.

## Materials and methods

### Trial oversight

The clinical trial was a Phase IIa, single-centre, open label, single-arm proof of concept study of a small number of patients, where the primary endpoint was safety and tolerability of subcutaneously administered ILB®. To assess possible efficacy in patients with ALS, the trial was conducted in a heterogeneous ALS patient group of intermediate disease severity. The study (EudraCT number 2017-005065-47) was conducted at the Sahlgrenska University Hospital, Gothenburg, Sweden. The trial was overseen and approved by the Ethics Committee of the University of Gothenburg and by the Swedish Medical Products Agency (reference number 21788). The trial was sponsored by Tikomed AB, who had no influence on the conduct of the trial and was not involved in data collection or analysis. The study protocol is described in S1 Appendix and the verbal and written information provided to the patients were in accordance with the Declaration of Helsinki. The underpinning data that support the findings in this study are available from the EU Clinical Trials Register (available from https://www. clinicaltrialsregister.eu/ctr-search/trial/2017-005065-47/results and on request from the authors).

### Investigational medicinal product (IMP), dose and mode of administration

The active pharmaceutical ingredient of the IMP is a unique and distinct low molecular weight dextran sulphate (LMW-DS) formulation, named ILB® whose structure, formulation, synthesis and structure has been previously described in detail in a published patent document (publication number: WO 2016/076780 –New dextran sulphate). ILB® was provided by Tikomed AB in 10 mL vials containing a solution of 20 mg/mL ILB® in 9 mg/mL NaCl. A single batch of drug was used throughout the study. ILB® was injected subcutaneously on alternating sides of the abdomen by the clinical personnel at the Sahlgrenska University Hospital. Five injections of 1 mg/kg, with a weekly dosing interval, were administered. The exact dose administered depended on the patient's body weight at Visit 2 (Day 1), prior to the first ILB® administration.

### Patients

The planned patient recruitment number to this safety and tolerability trial at the Sahlgrenska University Hospital was 15 patients with ALS. This report describes the accumulated data from the first 13 patients recruited into the clinical trial, that was halted early due to the drug's confirmed safety profile. Persons of both sexes with a definite diagnosis of ALS, including sporadic and genetic forms with either slow or rapid progression in the early phase of the disease, were screened and included into the study. The inclusion/exclusion criteria for trial entry are summarized in Table 1. Individuals were included in the drug trial after giving informed written consent if the diagnosis of ALS was confirmed as definite according to the El Escorial criteria [32], if there was no other major degenerative or inflammatory disease and if there was a

**Table 1. Trial inclusion and exclusion criteria.**

| Inclusion Criteria | Exclusion Criteria |
|---|---|
| • Willing and able to give written informed consent for participation in the study<br>• Definite clinical diagnosis of ALS<br>• Male or female patients between 18 to 80 years old (inclusive)<br>• Forced vital capacity >65% of predicted value for gender, height, and age at screening<br>• Evaluated with ALSFRS-R and Norris clinical rating scales for at least the past 4 weeks before study drug administration | • Unable to understand information about the study or were expected not to collaborate with the study team<br>• Concurrent serious disease, other than ALS, at the discretion of the Investigator<br>• Pregnancy: Patients of childbearing potential not willing to use adequate double contraception1 with<br>• <1% failure rate after the screening visit until the last visit<br>• Addiction to drugs or alcohol<br>• Confirmed HIV, hepatitis B or hepatitis C<br>• Known bleeding disorders or abnormal bleeding events<br>• Treatment with anticoagulant drugs warfarin and novel oral anticoagulants (NOAC) within 14 days prior to screening<br>• Treatment with riluzole or lamotrigine within 28 days prior to study drug administration<br>• Hypersensitivity to dextran sulphate<br>• Poor venous access<br>• Patients with clinically significant abnormal prothrombin complex-international normalised ratio (PK-INR), fibrinogen, von Willebrand factor and APTT at screening. |

ventilatory capacity of no less than 65% of normal predicted Forced Vital Capacity (FVC) at screening.

Motor neuron dysfunction and degeneration in ALS involves several pathogenic mechanisms, which include disturbed energy metabolism, cytoskeletal abnormalities, changes in transcription, glial hyperactivation and reduced glutamate uptake leading to glutamate excitotoxicity. Riluzole (Rilutek®) and lamotrigine are anti-glutaminergic drugs that are often prescribed to patients with ALS. These drugs act by reducing glutamate synthesis and release, thereby reducing glutamate excitotoxicity [12–14]. ILB® evaluated in this trial is able to modulate molecular processes relevant to ALS by enhancing the biological effects of specific growth factors which have multiple downstream consequences, including the enhancement of glutamate uptake by astrocytes [23, 24]. Specifically, in glial cells ILB® reduces oxidative stress, mitochondrial dysfunction and glial activation. It also promotes cell survival, differentiation, MBP expression and glutamate uptake. In neuronal cultures ILB® reduces neuronal cell death, oxidative stress and mitochondrial dysfunction. In neurons the compound also promotes differentiation, neurite outgrowth and cellular homeostasis. Most importantly glutamate production and release in neuronal cells is not affected by ILB®. Rather, it enhances glutamate uptake by astrocytes. Glutamate production and release (followed by $Ca^+$-mediated signalling cascades) in neuronal cells is one of the most important factors required for the synaptic remodelling necessary for regeneration and repair of neuronal networks affected in ALS (and indeed all neurodegenerative diseases). This process is controlled by the growth factors modulated by ILB® and their enhanced bioavailability should lead to functional benefit and clinical improvement in patients with neurodegenerative conditions.

Riluzole and lamotrigine act on several neurotransmitter systems and signalling pathways. *In vitro* studies and patient side effect profiles of the drugs demonstrate that they affect

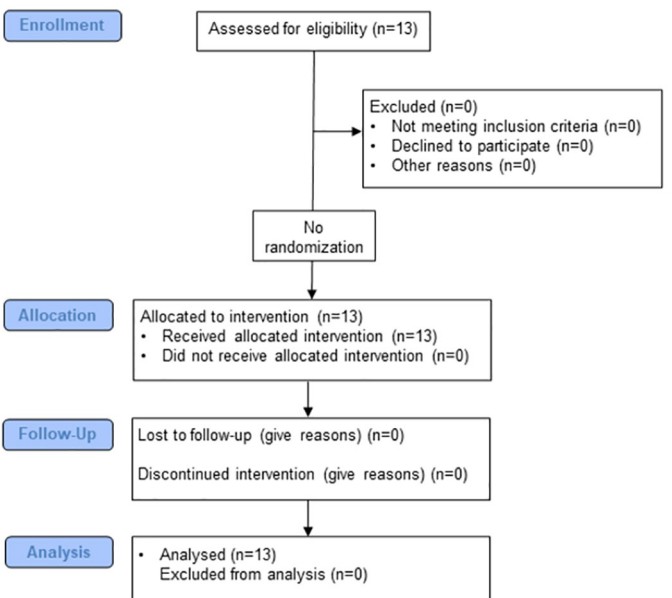

**Fig 1. Consort flow diagram for clinical trial (EudraCT number 2017-005065-47).**

glutamatergic signalling in neurons and most importantly inhibit $Ca^+$-dependent downstream signalling in these cells [12–14]. This means that both of these anti-glutamatergic drugs inhibit mechanisms that are pivotal for the neuronal repair induced by ILB® under test in this study. In summary, based on the established mechanism of action of ILB®, it is apparent that riluzole and lamotrigine will interfere with key ILB® actions and attenuate the beneficial effects in patients. Accordingly, patients on riluzole or lamotrigine were excluded from the trial due to suspected potential interactions with ILB®. Most patients were off these drugs before they volunteered for the trial. ILB® administration started only after a minimum of 28 days washout period. All patients were offered pharmacological treatment with established drugs indicated for ALS after the end of study. Double contraception (if applicable) was mandatory during the study.

## Trial design and schedule of events

The trial design and schedule of events are summarized in Fig 1 and Table 2. Whilst the primary endpoints of the study were drug safety and tolerability, secondary endpoints relating to drug efficacy and mechanism of action were also included. The absence of patient drop-out and any drug-related SAE were deemed to be the primary criteria for judging success in this study. After the initial screening visit (Visit 1a), patients had a lumbar puncture to sample CSF biomarkers (Visit 1b, a total CSF volume of 15 mL was taken). This was followed by 5 weekly dosing visits (Visits 2–6 on Days 1–29) for administration of a single subcutaneous (sc) ILB® injection per week of 1.0 mg/kg body weight in saline into the subcutaneous fat of the lower abdomen with a maximum of 1.5 mL at each injection site. Patients underwent another sampling of CSF (Visit 7 on Day 36) one week after the last ILB® injection. Further follow-up visits (Visits 8 and 9) were made at Day 50 and Day 99 after receiving the first ILB® dose, i.e., 3 and 10 weeks after the last ILB® sc injection. The functional disability of patients was assessed using the ALSFRS-R, Norris and autonomic/sensory symptom scores pre-treatment and

**Table 2. Trial design and schedule of events.**

| Visit | 1a | 1b | 2 | 3 | 4 | 5 | 6 | 7 | 8 | 9 |
|---|---|---|---|---|---|---|---|---|---|---|
| | Screening | Screening | Dosing | Dosing | Dosing | Dosing | Dosing | Follow-up | Follow-up | End of study follow-up |
| Day | -30 to -7 | -23 to -1 | 1 | 8 | 15 | 22 | 29 | 36 | 50 | 99 |
| Time window (days) | | | | ± 3 | ± 3 | ± 3 | ± 3 | ± 3 | ± 7 | ± 7 |
| Informed consent | X | | | | | | | | | |
| Information on contraception | X | | | | | | | | | |
| Information on medication | X | | | | | | | | | |
| Eligibility criteria | X | | | | | | | | | |
| Demographics | X | | | | | | | | | |
| Medical history | X | | | | | | | | | |
| Concomitant medications | X | X | X | X | X | X | X | X | X | X |
| Physical examination | X | | X | X | X | X | X | X | X | X |
| Weight | X | | X | | | | | X | | |
| Height | X | | | | | | | | | |
| Vital signs | X | | X | X | X | X | X | X | X | X |
| Forced vital capacity (FVC) | X | | X | X | X | X | X | X | X | X |
| Haematology, including PT- INR | X | | X | X | X | X | X | X | X | X |
| Clinical chemistry | X | | X | X | X | X | X | X | X | X |
| Pregnancy | X | | X | X | X | X | X | X | X | X |
| Drugs of abuse | X | | | | | | | | | |
| Gamma-glutamyltransferase | X | | | | | | | | | |
| HIV, hepatitis B and C | X | | | | | | | | | |
| ECG | X | | | | | | | X | | |
| ALSFRS-R and Norris rating scales | X | | X | X | X | X | X | X | X | X |
| Quality of life assessment | X | | X | | X | | | X | | |
| Autonomic/sensory symptoms | X | | X | X | X | X | X | X | X | X |
| Biomarkers (CSF, plasma, serum) | | X | | | | | | X | | |
| ILB® administration | | | X | X | X | X | X | | | |
| Blood sampling PK, HGF, APTT | | | X | | | | X | | | |
| Blood sampling for explorative analyses | | X | | | X | | | X | | |
| CSF sampling for exploratory analyses | | X | | | | | | X | | |
| Adverse events | X | X | X | X | X | X | X | X | X | X |

weekly thereafter for the 5 treatment weeks and at the follow-up visits. In parallel there was clinical evaluation of quality of life using a visual analogue scale and by dialogue between the clinical investigator and each patient plus their next of kin. Measurements of spirometry, body weight and collection of blood for laboratory safety and biomarker tests were made at each dosing visit. These measurements were repeated at 36, 50 and 99 days after treatment initiation (i.e., 7, 21 and 70 days after the last administered dose of ILB®).

## Outcome measures

**Pharmacokinetics of ILB® and HGF measurements.** Blood samples for pharmacokinetic (PK) analysis of ILB® and HGF plasma levels were collected through venipuncture or through an indwelling venous catheter into a vacutainer tube with citrate. Blood samples for sample dilution, generation of standard curves and baseline measurements were collected prior to dose administration. Collection of blood began at the start of the ILB® injection and continued at prescribed time intervals up to 6 hours. The post-injection kinetics of plasma

ILB® and HGF were measured using proprietary methods by Eurofins Biopharma Product Testing (Munich, Germany) and using ELISA by the Clinical Chemistry Laboratory at the Sahlgrenska University Hospital (Gothenburg, Sweden), respectively.

**Clinical safety and tolerance parameters.** Frequency, seriousness, and intensity of treatment-emergent adverse events (TEAE) or serious adverse events (SAE) were assessed throughout the study by patient reporting and testing/observation/questioning by medical personnel or investigators. An AE was defined as any untoward medical occurrence in a patient who had been administered ILB® and which did not necessarily have a causal relationship with this treatment. An AE could therefore be any unfavourable and unintended sign (including an abnormal laboratory finding), symptom, or disease temporally associated with the use of ILB®, whether or not related to the drug. A baseline event was defined as any AE in a patient that occurred after he/she signed the consent form up until the first administration of ILB®. A TEAE was defined as any AE that was not present prior to the initiation of ILB® administration or any event already present that worsened in either intensity or frequency following exposure to ILB®.

Blood tests were routinely collected at each visit during the diagnostic phase as well as during treatment and post-treatment visits to include assessment of organ function plus haematology and haemostatic parameters. All participants had a physical examination. Body weight and vital signs were recorded, pregnancy testing (if applicable), electrocardiogram, drug screening, spirometric investigation of Forced Vital Capacity (FVC), as well as a quality of life assessment, HIV and hepatitis serology tests, were carried out at selected visits (see Table 2 for the assessment timetable).

**CSF, blood, serum, and plasma sampling.** CSF, blood, serum, and plasma were collected for clinical and exploratory scientific use. CSF was collected by lumbar puncture (spinal level L3-L4; 22G spinal non-traumatic needle). The first lumbar puncture was performed after patient screening and inclusion but before the first administration of ILB®. The second lumbar puncture was performed one week after the last ILB® injection. Blood samples were drawn at defined study intervals by a venous catheter into vacutainer tubes. Laboratory analyses of blood, serum, plasma and CSF were performed immediately after collection by the Clinical Chemistry Laboratory at the Sahlgrenska University Hospital. Some CSF and plasma samples were collected and frozen immediately prior to storage (-80˚C) in a biobank in the Sahlgrenska University Hospital until analysis was performed with a complete sample set for the specific biomarker analysis.

**Laboratory blood tests for drug safety evaluation.** Analyses of sodium, potassium, chloride, calcium, albumin, aspartate aminotransferase, alanine aminotransferase, alkaline phosphatase, C-reactive protein, glucose, total bilirubin, haemoglobin, haemoglobin S, HbA1c, red blood cell counts, white blood cells, differential cell count and thrombocytes (platelets), fibrinogen, von Willebrands Factor, as well as the coagulation measures of prothrombin time (PT) together with the international normalized ratio (INR) and activated partial thromboplastin time (APTT), were executed during the study by the Clinical Chemistry Laboratory at Sahlgrenska University Hospital; the latter for safety reasons due to the potential for a discrete anticoagulant effect of the ILB®.

**Disease-related biomarker analysis.** Analysis of CSF for albumin, IgG, IgM, IgGindex, IgMindex, Tau- protein, phosphor-Tau protein, Neurofilament light chain (NfL) and proteasome-complement complex was performed during the study. Analysed biomarkers in serum included albumin, IgG and IgM. Biomarkers analysed in plasma included myoglobin, creatine kinase, NfL and proteasome-complement complex (compleasome). These analyses were all carried out by the Clinical Chemistry Laboratory at Sahlgrenska University Hospital.

**Functional rating of disability by ALSFRS-R and Norris rating scales.** The ALSFRS-R provided a physician-generated validated assessment of the patient's degree of functional impairment, which was evaluated serially to objectively assess any impact of treatment on the progression of disease [33]. The ALSFRS-R included questions that asked the physician to rate his/her impression of the patient's level of functional impairment in performing twelve common tasks. Each task was rated on a five-point scale from 0 = cannot do, to 4 = normal ability. Individual item scores were summed to produce a reported score of between 0 = worst and 48 = best. ALSFRS-R scores were analysed to provide data on the absolute change from baseline over 3 months. The Norris rating scale provided another physician-generated validated assessment of the patient's degree of functional impairment, which was evaluated serially also to objectively assess any response to treatment or progression of disease. The Norris rating scale included questions that ask the physician to rate his/her impression of the patient's level of functional impairment in performing 34 common tasks and bodily functions [34]. Each task or function was rated on a four-point scale from 0 = cannot do, to 3 = normal ability. Individual item scores were summed to produce a reported score of between 0 = worst and 100 = best. Norris scores were analysed to provide data on the absolute change from baseline over 3 months. The physician-rated assessments by ALSFRS-R and Norris scales were made after studies of functions at visits and questioning the patient about social function and disabilities.

**Subjective rating of non-motor, autonomic and sensory symptoms.** The non-motor, autonomic and sensory symptom rating scale provided a bespoke physician- generated estimate of the severity of patient's non-motor/autonomic/sensory symptoms, which could be evaluated serially to subjectively assess any response to treatment or progression of disease. The rating scale included questions that ask the physician to rate his/her impression of the patient's symptom severity relating to 16 non-motor, autonomic and sensory functions, including: pain; paresthesia; coldness; sweating; constipation; frequency of faeces; frequency of urination; urination difficulty, numbness; fatigue; tiredness; insomnia; arrhythmia; tachycardia; frequent waking; vertigo/dizziness. This patient cohort referred to 8/16 functions in their responses. Each symptom was rated on a five-point scale from 0 = none, to 4 = very severe. Total scores for autonomic/sensory symptoms provided data for comparison at Day 1, Day 36 and Day 99.

**Spirometry.** All patients were investigated by spirometry at each visit and Forced Vital Capacity (FVC) was measured as the best recording of three attempts at each visit.

**Quality of life (QoL).** This parameter was assessed by a visual analogue scale (VAS) that comprised a questionnaire with three question sets for patient and next-of-kin self-reporting relating to general, physical, and mental health status [35]. Each question set generated data on a scale from 0 = very bad, to 100 = very good. QoL scores were generated at intervals during the ILB® treatment period to provide data on the absolute change from baseline.

## Statistical analysis

The ethical and practical challenges of carrying out randomised placebo-controlled trials for this rare and fatal disease group meant that a small single arm trial design was deemed acceptable by the regulatory authorities for safety and tolerability assessment. This was an open label, single centre, Phase IIa proof of concept study with safety and tolerability as the primary trial outcome, so no formal sample size calculation was performed. The sample size was determined empirically and reflects the exploratory nature of the trial and the rarity of ALS. Validation of the planned (15 patients) and actual patient number (13 patients) for this exploratory clinical trial is described in detail in the S2 Appendix Supplementary Methods. The proposed sample

size was considered sufficient by the Ethics Committee of the University of Gothenburg and by the Swedish Medical Products Agency for an early Phase II exploratory trial to provide preliminary data on treatment-related adverse events and to observe trends for treatment effects on the efficacy measures chosen for this study (pilot proof of concept study with summary outcome measures, including mean, median, mode, minimum value, maximum value, range, standard deviation, etc.). Categorical data is presented as counts and percentages. Continuous data are summarised using descriptive statistics. The baseline pre-treatment measurements (Visit 2) of the patients were compared to those at intervals during and after treatment using either the paired Student $t$ test or analysis of variance as applicable. In quantitative data, values of $P<0.05$ were considered as statistically significant. There was no correction of the Type I error (multiple analysis) due to the exploratory nature of the study. Statistical analyses were performed using Statistica software and SAS systems (SAS version 9.4, SAS Institute Inc., Cary, NC, USA). All pharmacokinetic calculations were performed using Phoenix® WinNonlin version 8.1, build 8.1.0.3530 (Certara, Princeton, NJ, USA).

## Results

### Patient demographics and clinical characteristics

The planned patient recruitment number was 15 but the slow recruitment and confirmed drug safety parameters led to early trial termination. This report describes the accumulated data from the 13 patients actually recruited into the clinical trial at the Sahlgrenska University Hospital, whose demographics and baseline clinical characteristics are summarized in Tables 3 and 4. The expected ALSFRS-R score range was calculated using the GLM and pre-slope models [36]. The trial data is reported on-line at https://www.clinicaltrialsregister.eu/ctr-search/trial/2017-005065-47/results.

### Pharmacokinetics of ILB®

Table 5 shows post-injection mean plasma ILB® levels from the patient cohort after injection on Day 8 and Day 29. Fig 2 shows plasma levels in individual patients after Day 1 and Day 29 injections. Following sc injection of ILB®, $C_{max}$ was reached by 2.5 hours, so that plasma levels of ILB® were significantly and transiently elevated to a transient peak at pharmacologically relevant levels (e.g., $3.34 \pm 0.6$ µg/mL after the Day 29 injection).

**Table 3. Baseline patient demographics and clinical characteristics.**

|  |  | Patients recruited (N = 13) |
| --- | --- | --- |
| **Age (years)** | Mean | $56.5 \pm 13.3$ |
|  | Median (Min, Max) | 58 (31, 80) |
| **Body Mass Index (kg/m$^2$)** | Mean | $25.2 \pm 2.9$ |
|  | Median (Min, Max) | 25 (21, 31) |
| **Height (cm)** | Mean | $178.7 \pm 11$ |
|  | Median (Min, Max) | 179 (158, 193) |
| **Weight (kg)** | Mean | $80.6 \pm 11.6$ |
|  | Median (Min, Max) | 82.8 (65, 96) |
| **Gender** | Female | 3 (23%) |
|  | Male | 10 (77%) |
| **Race** | White | 13 (100%) |
| **Baseline ALSFRS-R Score** | Mean | $36.31 \pm 6.66$ |
| **Baseline Norris Score** | Mean | $70.61 \pm 13.91$ |

**Table 4. Patient characteristics.**

| Patient number | Years since disease symptom onset | ALSFRS-R at baseline | Decline rate at baseline (points/month) | Expected ALSFRS-R at 99 days (GLM-Preslope models) |
|---|---|---|---|---|
| 1 | 2 | 44 | 0.17 | 38–43 |
| 2 | 2 | 43 | 0.21 | 37–42 |
| 3 | 2 | 37 | 0.46 | 30–35 |
| 4 | 2 | 42 | 0.25 | 36–41 |
| 5 | 6 | 39 | 0.13 | 29–39 |
| 6 | 1 | 44 | 0.33 | 39–43 |
| 7 | 7 | 26 | 0.26 | 14–25 |
| 8 | 6 | 33 | 0.21 | 23–32 |
| 9 | 2 | 37 | 0.45 | 30–35 |
| 10 | 8 | 22 | 0.27 | 9–21 |
| 11 | 1 | 34 | 1.17 | 27–30 |
| 12 | 1.5 | 35 | 0.72 | 28–33 |
| 13 | 2 | 34 | 0.58 | 27–32 |

## Plasma HGF levels

By 2.5 hours after each sc ILB® injection, plasma levels of HGF were significantly elevated around 40-fold from pre-treatment levels (Table 5), indicating a rapid trophic response to ILB® treatment. The dramatic rise in plasma HGF levels after ILB® sc injection (e.g., from 827 ± 598 to a peak of 32,438 ± 5,348 pg/mL at 2.5 hours after the Day 8 injection) was followed by a similarly rapid decrease towards baseline by 6 hours. Fig 3 shows the change in plasma HGF levels in individual patients after the Day 29 injection.

## Adverse events

There were no deaths or SAEs reported during the study. There was no discontinuation of IMP administration during the study and no patient withdrawal due to AEs. During the treatment period, 9/13 patients experienced 14 treatment emergent adverse events (TEAE), with 4 of these judged as possibly related to the IMP (acne, subcutaneous hematoma, fatigue, and pyrexia) (Table 6). These TEAEs were of mild or moderate intensity and were all resolved without requiring any action related to the IMP. All injection-related bruises healed within one week without discomfort to the patient. Lumbar puncture at spinal level of L3-L4 was performed twice in each patient (one week before first and one week after last ILB® dose) and left no haematoma or remaining pain. One patient had a short-lasting pain during spinal tap of the first puncture but not on the second occasion. One patient had a moderate headache during the day after lumbar puncture, resolving within 24 hours without treatment. Two patients with need of a walking aid due to their ALS symptoms stumbled and fell during walking (one while travelling to the clinic, the other at home) during the study and got minor skin bruises,

**Table 5. Plasma ILB® and Hepatocyte Growth Factor (HGF) levels at Day 8 and Day 29 injections (N = 13; Mean ± SD).**

| Sample time | Plasma ILB® (µg/mL) Day 8 | Plasma ILB® (µg/mL) Day 29 | Plasma HGF (pg/mL) Day 8 | Plasma HGF (pg/mL) Day 29 |
|---|---|---|---|---|
| **Pre-injection** | Below detection | Below detection | 827 ± 598 | 724 ±223 |
| **Maximum peak 2.5 hours post-injection** | 3.16 ± 0.61 | 3.34 ± 0.6 | 32,438 ± 5,348 (p<0.001 *vs* pre-injection) | 41,691 ± 8,005 (p<0.01 *vs* pre-injection) (p = 0.01 *vs* Day 8) |
| **6 hours post injection** | 1.27 ± 0.77 | 2.28 ± 0.62 | 10,687 ± 5,795 | 6,423 ± 3,370 (p = 0.022 *vs* Day 8) |

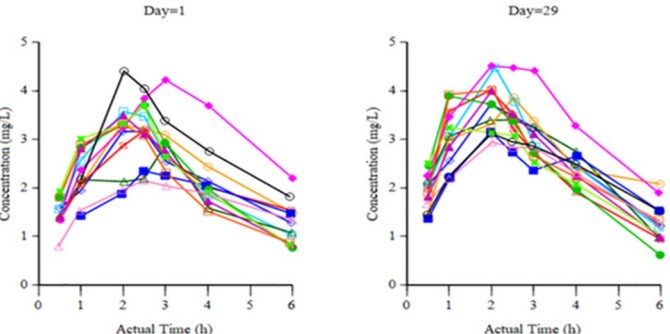

**Fig 2. Post-injection plasma ILB® levels for individual patients at Day 1 and Day 29.**

healing without scars. Both had a history of stumbling and falling before and after the clinical study (for further details see https://www.clinicaltrialsregister.eu/ctr-search/trial/2017-005065-47/results). The treatment was well tolerated throughout with no patient withdrawals from the study.

## Blood analyses for drug safety

None of the tests investigating organ function and haematology biomarkers in serum, plasma or blood showed any significant deviation from expected levels in ALS at any time point measured throughout the trial, including during or after ILB® administration (not shown). Kinetic analysis of activated partial thromboplastin time (APTT) showed an average time to coagulation before treatment of 26.5 secs, a top APTT time of 33.4 secs at 2.5 hours after ILB® injection followed by a decrease of APTT to 29.3 secs at 6 hours after ILB® injection

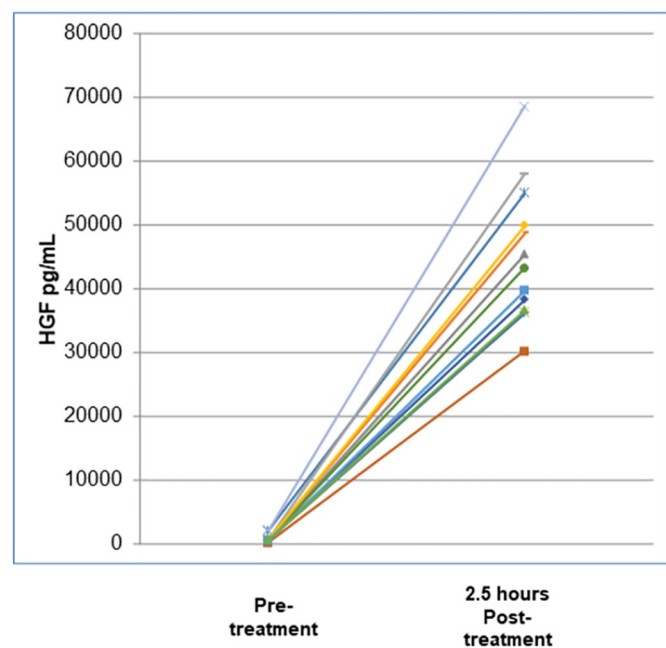

**Fig 3. Post-injection changes in plasma HGF levels in individual patients after the Day 29 injection of ILB®.**

**Table 6. Adverse event data (TEAE = treatment related adverse events; SAE = serious adverse events; N = Number of patients; M = number of events).**

| | ILB® treatment period (N = 13) | | Follow-up period (N = 13) | | Total (N = 13) | |
|---|---|---|---|---|---|---|
| | N (%) | M | N (%) | M | N (%) | M |
| Any TEAE | 9 (69%) | 14 | 6 (46%) | 8 | 11 (85%) | 22 |
| Any SAE | 0 | 0 | 0 | 0 | 0 | 0 |
| Any TEAE leading to withdrawal | 0 | 0 | 0 | 0 | 0 | 0 |
| Any TEAE leading to death | 0 | 0 | 0 | 0 | 0 | 0 |
| **Causality** Possibly Related Unrelated | 2 (15%) | 4 | 1 (8%) | 1 | 3 (23%) | 5 |
| | 8 (62%) | 10 | 6 (46%) | 7 | 11 (85%) | 17 |
| **Severity** Mild Moderate | 8 (62%) | 13 | 4 (31%) | 4 | 10 (77%) | 17 |
| | 1 (8%) | 1 | 4 (31%) | 4 | 4 (31%) | 5 |

(Fig 4). Thereafter, the APTT levels normalized spontaneously without measurable clinical symptoms.

### Functional disability measures (ALSFRS-R and Norris rating scales)

The patient group showed an improvement in the clinical ratings evidenced by both the ALSFRS-R and the Norris rating scales that became apparent after the first injection (Visit 2 at Day 1) and was enhanced throughout the treatment period until termination of the injections (Fig 5A and 5B). The changes in clinical scores from baseline illustrates the improvement in function experienced during ILB® treatment measured by both ALSFRS-R and Norris rating (e.g., an increase of +2.5 in the ALSFRS-R scores and +7.3 in the Norris scores over the treatment period). In addition, most patients reported an increased vitality (feeling more energetic, strong, and active) during the first day post-injection, with decreased spasticity and muscular weakness from Day 3 onwards associated with a concomitant improvement in muscular and bulbar symptoms. During the post-treatment follow-up visits, the measurable improvement in clinical function was maintained for 3–4 weeks after the last dosage, i.e., until Visit 8 at 50 Days. However, after this time the improvement in function was lost, so that the measures taken at Visit 9 at 99 Days, were not significantly different from the pre-treatment clinical ratings. However, none of the patients progressed to their predicted level of functional deficit based on their baseline characteristics by Day 36, and the attenuation of predicted disease

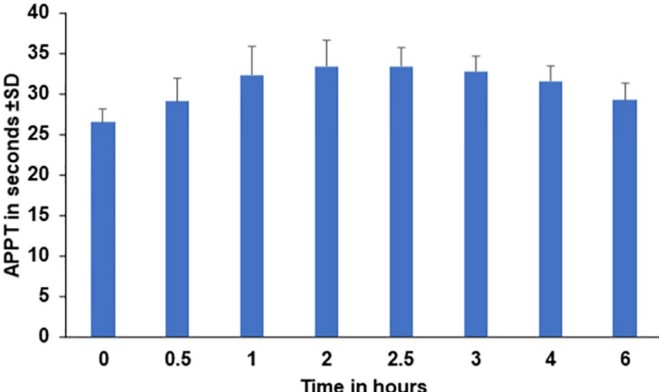

**Fig 4. Post-injection changes in activated partial thromboplastin time (APTT) after the Day 29 injection of ILB®.**

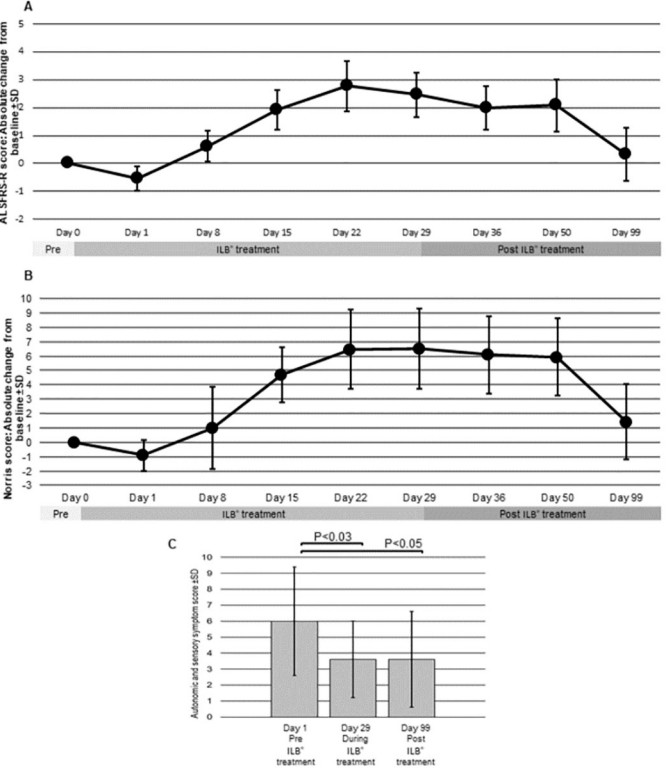

**Fig 5. Changes in the (A) ALSFRS-R, (B) Norris and (C) autonomic/sensory symptom scores from baseline (N = 13; Mean ± SD).** A rising ALSFRS-R and Norris score indicates functional improvement. Conversely, a falling autonomic/sensory symptom score indicates symptom improvement.

progression was sustained in most patients at Day 99 (Table 7). This indicated a lasting post-treatment beneficial effect of the ILB® injections of 3–4 weeks. While disease progression resumes after treatment cessation, the effect appears maintained over the next 7 weeks.

**Table 7. Comparison of actual and expected (predicted by GLM and preslope models) ALSFRS-R scores for individual patients at Visit 2 (V2 at Day 1; prior to first ILB® injection), Visit 7 (V7 at Day 36; 7 days after last ILB® injection) and Visit 10 (V10 at Day 99; 70 days after last ILB® injection); (N/A = data not available).**

| Patient number | ALSFRS-R at baseline | Actual ALSFRS at 36 days | Expected (GLM-Preslope) ALSFRS at 36 days | Actual ALSFRS at 99 days | Expected (GLM-Preslope) ALSFRS at 99 days |
|---|---|---|---|---|---|
| 1 | 44 | 44 | 39–44 | 45 | 38–43 |
| 2 | 43 | 43 | 38–43 | 43 | 37–42 |
| 3 | 37 | 40 | 32–36 | 40 | 30–35 |
| 4 | 42 | 45 | 37–42 | 37 | 36–41 |
| 5 | 39 | 42 | 30–39 | 39 | 29–39 |
| 6 | 44 | 46 | 40–44 | 39 | 39–43 |
| 7 | 26 | 32 | 16–26 | 27 | 14–25 |
| 8 | 33 | 35 | 24–33 | 38 | 23–32 |
| 9 | 37 | 43 | 32–36 | 41 | 30–35 |
| 10 | 22 | 23 | 10–22 | 24 | 9–21 |
| 11 | 34 | 38 | 28–33 | N/A | 27–30 |
| 12 | 35 | 38 | 29–34 | 38 | 28–33 |
| 13 | 34 | 34 | 28–33 | 33 | 27–32 |

**Table 8. Autonomic and sensory scores for individual patients.**

| Patient | Day 1 Visit 2 Pre-treatment | Day 36 Visit 7 During treatment | Day 99 Visit 9 Post-treatment | Complaints |
|---|---|---|---|---|
| 1 | 4 | 1 | 1 | Coldness, fatigue, paresthesia |
| 2 | 10 | 7 | 5 | Coldness, constipation, fatigue, tiredness |
| 3 | 4 | 3 | 0 | Fatigue, tiredness |
| 4 | 10 | 5 | 4 | Coldness, fatigue |
| 5 | 6 | 3 | 8 | Coldness, fatigue |
| 6 | 0 | 1 | 4 | Fatigue, constipation |
| 7 | 6 | 2 | 2 | Coldness |
| 8 | 2 | 0 | 0 | Coldness, frequent wake-ups |
| 9 | 3 | 2 | 8 | Coldness, fatigue |
| 10 | 10 | 4 | 2 | Coldness, frequent wake-ups |
| 11 | 5 | 6 | 0 | Fatigue |
| 12 | 12 | 7 | 7 | Coldness, fatigue, frequent wake-ups, paresthesia |
| 13 | 6 | 6 | 6 | Fatigue, sweating, vertigo |

## Clinical rating of autonomic and sensory symptoms

The scores for severity of the autonomic and sensory symptoms experienced declined significantly during the ILB® treatment period (Fig 5C and Table 8). In addition, most patients maintained their autonomic/sensory symptom improvement post-treatment up to the Day 99 assessment point.

## Spirometry

All subjects had a forced vital capacity (FVC) of > 65% of predicted normal FVC at enrolment. There was a 10% decrease in FVC from study entry at Day 1 to Day 36 (p<0.02, paired T-test), but no further significant change in FVC was observed during the 49 days between Visit 8 (Day 50) and Visit 9 (Day 99). The average weekly decrease in spirometry score during Days 1–36 was -3.36% per week, but this rate of decline was arrested by Day 36 and maintained at -0.49% per week over Days 36–99 (Table 9).

**Table 9. Clinical and laboratory measurements of patients with ALS (N = 13; Mean±SD; ns = not significant; nd = not determined; * = evidence of subgroup responsiveness).**

| Assessment | Day 1 Before treatment | Day 36 Post treatment | Day 99 Post treatment |
|---|---|---|---|
| *Clinical ratings* | | | |
| **Norris rating** | 70.61 ± 13.91 | 77.85 ± 14.24 (p = 0.028 *vs* Day 1) | 71.50 ± 12.31 (p = 0.028 *vs* Day 36) |
| **ALSFRS-R rating** | 36.31 ± 6.66 | 38.77 ± 6.44 (p = 0.008 *vs* Day 1) | 40.13 ± 6.01 (p = 0.027 *vs* Day 36) |
| **Autonomic/sensory symptom rating** | 6.00 ± 3.58 | 3.62 ± 2.40 (p = 0.003 *vs* Day 1) | 3.62 ± 3.01 (p = 0.052 *vs* Day 1) |
| *Quality of life scores* | | | |
| **Physical well-being** | 35.50 ± 23.54 | 39.88 ± 19.96 (p = 0.04 *vs* Day 1) | nd |
| *Spirometry* | | | |
| **FVC (% predicted normal FVC)** | 87.53 ± 13.51 | 80.46 ± 13.21 (p = 0.01 *vs* Day 1) | 76.01 ± 0.13 (p = 0.01 *vs* Day 1) (p = 0.04 *vs* Day 36) |
| *Plasma muscle biomarkers* | | | |
| **Plasma Myoglobin (µg/L)** | 133.92 ± 126.28 | 103.69 ± 72.16 (p = 0.021 *vs* Day 1) | nd |
| **Plasma Creatine Kinase (U/L)** | 421 ± 337 | 364 ± 299 (p<0.05 *vs* Day 1) | nd |

### Quality of life

There were no significant changes in general or psychological well-being scores (not shown); however, a small but statistically significant improvement in physical well-being was reported (Table 9).

### Disease biomarkers

The levels of CSF and serum NfL were stable during the study period, as were the levels of other disease biomarkers including tau, phosphor-tau, albumin, IgG, IgM, IgGindex, IgMindex and proteasome-complement complexes (for further details see https://www.clinicaltrialsregister.eu/ctr-search/trial/2017-005065-47/results). However, there was a statistically significant decrease from 421 ± 337 to 364 ± 299U/L (P<0.05) in plasma creatine kinase and also in plasma myoglobin (down from 133.92 ± 126, to 103.69 ± 72.16 μg/mL, P = 0.021) during the treatment period, indicating an attenuation in the rate of muscle tissue degeneration (Table 9).

## Discussion

The ILB® used in this study has now been designated an orphan medicinal product for ALS (an orphan disease) by both the European Medicine Agency and the Food and Drug Administration (United States). Small, single arm, open label trials with safety and clinically relevant endpoints are accepted as an essential first step in drug development for these challenging conditions [37]. In designing this first safety and tolerability clinical trial of ILB® in patients with ALS, we used well documented safety data, longitudinal data gathering, multiple endpoints, and baseline adjustments to design a trial that would generate meaningful proof of concept data from a relatively small cohort of participants.

Most importantly, a favourable safety and tolerability profile was evident for the ILB® treatment regime applied in this ALS patient cohort. There were no deaths or serious adverse events reported after ILB® administration, nor were there any patient withdrawals during the study. Of the 14 laboratory or patient reported treatment-emergent adverse events in this study, 10 were deemed to be unrelated to the IMP, with 13 being of mild and just 1 of moderate severity. This confirmed the positive safety profile seen with this drug in pre-clinical toxicology and Phase I/II human studies, even when several-fold higher doses of ILB® were used in these latter studies. Thus, the trial was implemented successfully to its planned conclusion, and the drug safety and tolerability revealed was such that larger efficacy trials in patients with ALS can be contemplated. By these criteria, the trial was judged to be a success.

The pharmacokinetics of ILB® indicated rapid blood absorption of the drug after sc ILB® injection, followed by fast clearance. As ILB® is excreted within 8–12 hours after intravenous injection [24], the clinical effects seen in the first week after injection were probably caused by the drug activating multiple biological mechanisms that were sustained after the elimination of ILB® from body fluids. It is possible that, like some other dextran sulphates, ILB® acts as a heparin mimetic, releasing and activating numerous heparin-binding growth factors sequestered in the extracellular matrix, aiding their redistribution and prolonging their half-life in soluble form [38, 39]. The release, activation and circulation of multiple growth factors from peripheral and central tissues would initiate a programme of diverse down-stream cellular responses relating to metabolism, tissue repair and regeneration. The PK of ILB® after sc injection in patients with ALS is strikingly similar to that seen in previous clinical studies [24] and with plasma HGF levels, both peaking at 2.5 hours. The parallel pharmacokinetics of ILB® and HGF suggests a heparinoid-like effect for ILB®, initiating the release of growth factor from the endothelial cell surface into the circulation.

Although the small size and open-label nature of this study, as well as the lack of rigour of the statistical analysis, necessitates an extremely cautious interpretation, the results of the secondary endpoints (including a limited panel of blood biomarkers that are reported here and elsewhere [40]) suggest the potential for treatment-related change in the progression of ALS. We are mindful that this trial was not placebo controlled, and open label trials are bound to have strong placebo effects in up to 30% of patients [41]. A measurable (using two widely adopted ALSFRS-R and Norris scoring systems) functional recovery emerged in this small patient cohort within one week of treatment initiation, that increased to statistical significance during the treatment period of five weekly ILB® injections. The alleviation of clinical symptoms lasted for 3–4 weeks after the last injection and was followed by a slow resumption of disease progression during the remaining 7 weeks of the follow-up period. Patient reported side effects of the drug were mild, with a feeling of increased physical vitality (defined as relating to the patient perceived levels of energy and fatigue) experienced 5–6 hours after ILB® injection, effects that lasted for the trial duration. This observation was reflected in a decrease after treatment of 25% (P<0.02) in the scores for patient perceived levels of tiredness and fatigue as measured in the Subjective Sensory and Autonomic Symptoms assessment panel. These reported rapid responses in patient perceived energy levels may relate, in part, to ILB® initiating improvements in tissue energy metabolism, as described elsewhere for this drug both in the injured rodent brain [23] and in this same cohort of patients with ALS [40].

The plasma levels of creatine kinase and myoglobin (biomarkers mainly related to degeneration/injury of muscle mass) were higher at entry to the study in the patients with ALS, as in a large proportion of patients with ALS, compared to the normal level in the population [42]. The levels of both plasma creatine kinase and myoglobin decreased significantly during ILB® treatment, with plasma myoglobin proving to be a more responsive measure than creatine kinase. This apparent inconsistency probably reflects the relatively short period of treatment and observation in this study, coupled with the earlier responsiveness, shorter half-life and availability of a more sensitive assay for plasma myoglobin [43]. However, the biochemical evidence of reduced muscle degeneration supports the clinical observations of improved muscle function reflected in the ALSFRS-R and Norris rating scales. Many of the functions relevant to these rating scores are mediated by cervical, trunk, lumbosacral, and respiratory muscles and scores in these categories show close agreement with objective measures of muscle strength [30]. The reduction in the rate of muscle atrophy evidenced by reduced plasma creatine kinase and myoglobin may also be linked to the observed post-treatment attenuation in the rate of pulmonary function decline.

ILB® treatment also decreased the scored symptoms of autonomic/sensory dysfunction and the common general complaint among the patients of coldness and freezing almost totally disappeared during the treatment course. With weekly injections of ILB®, patients reported decreased spasticity and improvements in gait, talking and swallowing, presumably reflecting the additional influence of mobilized neurotrophic and myotrophic factors. The improvements noted in motor and autonomic functions were reflected in the enhanced physical well-being score in the QoL assessment.

The indications of early functional benefit of ILB® treatment to neural and muscle tissues seen in this study may partly relate to an acute stimulant effect of ILB® on the release of heparin-binding growth factors, including HGF [38]. HGF is a potent endogenous neurotrophic and myogenic factor synthesised locally and stored in the extracellular matrix of peripheral and central tissues. It has direct protective activities on motor neurons and muscle cells and indirect activities through actions on macrophages and glia, by stimulating glucose transport and metabolism and also by reducing inflammation, oxidative stress and glutamatergic neurotoxicity [26–28, 44, 45].

How ILB® acts on the brain in patients with ALS remains to be established, although its reported positive effects on compromised brain energy metabolism [23, 40] would be important [46]. CSF NfL levels have been linked to the destruction of large CNS neurons and we have previously reported a direct relation between CSF levels of NfL and the aggressiveness of the disease, as revealed by the time of survival of the patient from first clinical symptoms of ALS until death [4]. Here, no change in CSF levels of NfL before and after ILB® treatment was evident. Since the turnover of CSF NfL is slow, the 6-week interval between lumbar punctures may have been too short for ILB® treatment to have a measurable impact upon the level of NfL in the CSF compartment. However, the stabilisation of this biomarker of neurodegeneration may be indicative of treatment benefit.

Neuroinflammation is a hallmark of ALS, particularly contributing to the peripheral neurodegeneration leading to muscle atrophy. Proteasomes are essential for the degradation of proteins in degenerating tissues and they form complexes with complement activation fragments (compleasomes) as part of the innate immune system response [17]. In this cohort of patients with ALS, CSF and plasma proteasome-complement complex levels were elevated prior to treatment compared to levels in healthy persons [47, 48], but there were no consistent treatment-related changes in the proteasome-complement levels in the patient group as a whole. Yet, patients with rapid progress of the disease had particularly high levels of plasma proteasome-complement complex at the start of ILB® treatment and this sub-group experienced a marked decrease (-60% of the initial very high levels of proteasome complex) after ILB® treatment.

## Conclusions

In this Phase IIa pilot clinical trial in patients with ALS, ILB® treatment was shown to be safe and well tolerated. There were few side-effects, no severe adverse events and good drug tolerance after injection of ILB® at the tested dose once a week for 5 weeks. The increased functional scores after treatment initiation associated with stabilisation of most disease biomarkers must be cautiously interpreted due to the significant study limitations (small size and open-label nature of this study, lack of rigour of statistical analysis, etc.) but are suggestive of a possible disease modifying effect. Further studies on ILB® dosage and duration, inter-dosage intervals and possible long-term effects are in progress, but the present report supports ILB® as the first safe, well tolerated treatment with potential to arrest/reverse the clinical symptoms of ALS.

## Supporting information

**S1 Appendix. ILB® clinical trial protocol.**
(DOCX)

**S2 Appendix. Supplementary methods.**
(DOCX)

**S1 Checklist. TREND checklist.**
(TIFF)

## Acknowledgments

The authors would like to thank the patients participating in the study as well as their clinical and nursing staff at the Sahlgrenska University Hospital. In addition, the authors acknowledge

the contribution of Bernardo M. Ropero, Kaj Blennow, Ewa Johansson, Stefan Lange of the Sahlgrenska Academy, University of Gothenburg, for help with data collection.

## Author Contributions

**Conceptualization:** Ann Logan, Lars Bruce, Lennart I. Persson.

**Data curation:** Valentina Di Pietro, Barbara Tavazzi, Giuseppe Lazzarino, Giacomo Lazzarino, Lennart I. Persson.

**Formal analysis:** Ann Logan, Zsuzsanna Nagy, Nicholas M. Barnes, Antonio Belli, Valentina Di Pietro, Barbara Tavazzi, Giuseppe Lazzarino, Lennart I. Persson.

**Funding acquisition:** Lennart I. Persson.

**Investigation:** Lennart I. Persson.

**Methodology:** Ann Logan, Lennart I. Persson.

**Project administration:** Lennart I. Persson.

**Supervision:** Lennart I. Persson.

**Validation:** Lennart I. Persson.

**Writing – original draft:** Ann Logan, Lennart I. Persson.

**Writing – review & editing:** Ann Logan, Zsuzsanna Nagy, Nicholas M. Barnes, Antonio Belli, Valentina Di Pietro, Barbara Tavazzi, Giuseppe Lazzarino, Giacomo Lazzarino, Lars Bruce, Lennart I. Persson.

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
