## [Decision Letter · Decision Letter 0]

24 May 2021

PONE-D-21-04468

A phase II open label clinical study of the safety, tolerability and efficacy of a low molecular weight dextran sulphate for Amyotrophic Lateral Sclerosis

PLOS ONE

Dear Dr. Logan,

Thank you for submitting your manuscript to PLOS ONE. After careful consideration, we feel that it has merit but does not fully meet PLOS ONE’s publication criteria as it currently stands. Therefore, we invite you to submit a revised version of the manuscript that addresses the points raised during the review process.

We look forward to receiving your revised manuscript.

Kind regards,

Tai-Heng Chen, M.D.

Academic Editor

PLOS ONE

"Patents pertaining to this LMW-DS drug have been filed by Tikomed AB. LB is co-inventor of LMW-DS and a board member of Tikomed AB. AL, ZN and NMB declare consultancy payments from Tikomed AB and/or Axolotl Consulting Ltd for services outside the submitted work. The other authors declare that they have no competing interests."

We note that one or more of the authors have an affiliation to the commercial funders of this research study : Tikomed AB, Axolotl Consulting Ltd.

2.1. Please provide an amended Funding Statement declaring this commercial affiliation, as well as a statement regarding the Role of Funders in your study. If the funding organization did not play a role in the study design, data collection and analysis, decision to publish, or preparation of the manuscript and only provided financial support in the form of authors' salaries and/or research materials, please review your statements relating to the author contributions, and ensure you have specifically and accurately indicated the role(s) that these authors had in your study. You can update author roles in the Author Contributions section of the online submission form.

2.2. Please also provide an updated Competing Interests Statement declaring this commercial affiliation along with any other relevant declarations relating to employment, consultancy, patents, products in development, or marketed products, etc.  

2.3. We note that you have a patent relating to material pertinent to this article. Please provide an amended statement of Competing Interests to declare this patent (with details including name and number), along with any other relevant declarations relating to employment, consultancy, patents, products in development or modified products etc. Please confirm that this does not alter your adherence to all PLOS ONE policies on sharing data and materials, as detailed online in our guide for authors http://journals.plos.org/plosone/s/competing-interests by including the following statement: "This does not alter our adherence to  PLOS ONE policies on sharing data and materials.” If there are restrictions on sharing of data and/or materials, please state these. Please note that we cannot proceed with consideration of your article until this information has been declared.

Reviewers' comments:

Reviewer's Responses to Questions

**Comments to the Author**

1. Is the manuscript technically sound, and do the data support the conclusions?

Reviewer #1: Yes

Reviewer #2: Partly

Reviewer #3: Yes

Reviewer #4: Partly

2. Has the statistical analysis been performed appropriately and rigorously? 

Reviewer #1: Yes

Reviewer #2: No

Reviewer #3: I Don't Know

Reviewer #4: Yes

3. Have the authors made all data underlying the findings in their manuscript fully available?

Reviewer #1: Yes

Reviewer #2: No

Reviewer #3: Yes

Reviewer #4: Yes

4. Is the manuscript presented in an intelligible fashion and written in standard English?

Reviewer #1: Yes

Reviewer #2: Yes

Reviewer #3: Yes

Reviewer #4: Yes

5. Review Comments to the Author

Reviewer #1: Ann Logan et al.

Amyotrophic lateral sclerosis (ALS) is a neurodegenerative disorder characterised by rapid progression of upper and lower motor neuron loss, which causes paralysis and death within 2-5 years from disease onset. However only riluzole, which have beneficial effects in a limited population of patients with ALS, have been approved.

In this manuscript, Ann Logan et al., revealed that no serious adverse events or participant withdrawal were observed in clinical trial of LMW-DS in patients with ALS. The results also showed that ALSFRS-R improved after treatment. Although this study provides important results, this paper will be improved by addressing these issues. The specific concerns are listed below.

Major issues:

1. Progression rate of ALSFRS-R and disease duration should be included in the background of patients, as the change in ALSFRS-R is strongly influenced by the disease progression.

2. Effectiveness of treatment should be established by the demonstration of a treatment effect on function in daily activities such as ALFRS-R. However, given the open-label trial, it would have been helpful to include quantitative motor function as an objective measurement.

3. It is not a scientific expression to state that a patient's vitality has increased, especially in an open-label trial.

4. FDA recommends the consideration of add-on designs, in which a treatment previously shown to be effective for the treatment of ALS is given to all patients participating in the trial, with patients randomized to the added investigational drug or added placebo. Too much justification for open-label trial should be avoided. In addition, it should be explained in detail that patients taking riluzole are not eligible to participate in this trial.

5. Pain and autonomic dysfunction are not typical neurological findings in ALS patients, why did you assess these symptoms?

6. I can't find out the details of the scores for severity of the autonomic and sensory symptoms, but are vertigo and fatigue classified autonomic or sensory symptoms?

7. To investigate the correlation between the change of serum HGF and clinical measures such as ALSFRS-R would support the hypothesis that increased serum HGF improves motor function of patients with ALS.

Minor issues

1. What subscores of the ALSFRS-R have improved in ALS patients?

2. HGF should be spelled out.

3. The authors need to correct writing errors (FCV→FVC, Line226; forced ventilatory capacity→forced vital capacity, Line294).

Reviewer #2: This is a manuscript reporting the outcome in an early Phase 2 trial. As it stands there are serious issues in the reporting as the paper does not reflect the protocol. For example, the primary endpoints of the protocol are not mentioned here; nor are the success criteria - what is considered acceptable on these endpoints?

Also the trial was slated to recruit 15 patients. Only 13 were apparently approached according to the flow chart (not a CONSORT diagram as the trial is not randomised). So why was the number to be approached changed? Why was the trial not conducted as stated in the protocol. The methods section does not state the aspiration of the study, and the actual planned number is hidden in the protocol.

The patient characteristics are a result and not a method - this section including Table 1 needs to lead the results.

Please explain why the use of statistical tests here is not in line with the protocol which talks of summary measures only. Please give actual p-values and not just p<0.xx when the p-value exceeds 0.001. What work was done to demonstrate that the data were Normally distributed?

It is incorrect to only report significant results (e.g. Table 7) - please give effect sizes and CI here and report all tests done - what was the hypothesis here?

Dynamite plots (e.g. Fig 4,5) give less information than a table would - especially given the serial nature of the data Figure 5C is in particular the wrong figure and again selects only the significant differences. Figure 6 would be clearer as a table.

Reviewer #3: This is a concept-of-proof open, uncontrolled study on a very small and clinically heterogeneous ALS population treated with LMW-DS and followed up for a short time. The molecule is supposed to be effective mainly by stimulating the production of HGF.

The molecule and hypothesized mechanism of action is quite innovative and interesting. In addition, new trials in this population are always welcome given the lack of effective treatments for such a severe and incurable condition. Nevertheless, preclinical data about LMW-DS in neurological disorders are limited and experiments on ALS animal models are lacking. This limitation must be declared by the authors.

Safety and tolerability as primary endpoint are well documented and particular attention has been paid to any coagulation disorder potentially related to HGF. In my view, nevertheless, no inference can be drawn about clinical advantages. A clinical impression of "increased vitality, decreased spasticity and increased mobility" and even a transient improvement in functional scales have no relevance in such a small uncontrolled trial. Only robust changes in biological markers can be taken into account in this phase of the experimental study. This strong limitation must be stressed by the authors.

Despite these limitations, in my opinion, the study deserves attention and additional development

Reviewer #4: This manuscript by Logan and Coll. reports results of a phase II clinical study with molecular weight dextran sulphate in ALS. Besides demonstrating a good safety and tolerability of the IMP (primary end-point of the study), authors noted a significant clinical benefit of the drug on patients' mobility (as measured according to the ALSFRSr and Norris rating).

Basically flawn by the very small number of enorolled patients, this study raises questions and concerns.

Questions:

1. ALSFRSr score improves on average of 4 points after 3 months since the first injection of the IMP (and after 2 months after the last injection). Then, how do the authors explain the worsening of FVC values at day 99? Which are the ALSFRSr items improving?

2. What is the reference interval for CK values? why there was no blood CK sampling at day 99? Are creatinine values available?

3. why did the authors evaluate autonomic functions? What exactly is the autonomic and sensory rating scale used in the trial?

Concerns:

1. the IMP used in this study seems to reverse the natural decline of ALSFRSr overtime observed in ALS (without affecting objective measures like NFl or FVC). It is established that NFls are stable throughout the disease, therefore NFL stabilization shown here was expected and it is not a "positive" result. Furthermore, benefit is prompt. Authors should discuss, somewhere in the discussion, the chance of a placebo effect

2. Accordingly, why did not the authors evaluate QoL at day 99?

6. PLOS authors have the option to publish the peer review history of their article (what does this mean?). If published, this will include your full peer review and any attached files.

Reviewer #1: **Yes: **Shinichiro Yamada

Reviewer #2: No

Reviewer #3: No

Reviewer #4: No

---

## [Author Response · Author response to Decision Letter 0]

19 Jul 2021

Ann Logan et al. A phase II open label clinical study of the safety, tolerability and efficacy of ILB® for Amyotrophic Lateral Sclerosis

Response to Editor and Reviewers

We appreciate the detailed and helpful comments of the reviewers that we have considered carefully, and we believe that the revisions we have made in response have improved the manuscript significantly.

Please note that we have replaced the Recommended International Non-Proprietary Name (rINN) for the drug under test with its proprietary name and defined the rINN in the abstract, introduction and methods sections.

Editor

Major issues:

1. Progression rate of ALSFRS-R and disease duration should be included in the background of patients, as the change in ALSFRS-R is strongly influenced by the disease progression. 

Response: The Lead Clinical Investigator for this trial has treated more than 1000 ALS patients during his 45 years as a neurologist, following patients from diagnosis until death and has made multiple studies both on drug treatment as well as on the course of the disease. In Scandinavia there are at least 30 different ALS forms, by genetic, inflammatory and/or destructive kind. ALS is thus a very heterogeneous constellation of disease. ALS causes a degeneration of neurons and glia. The destruction of cells and the progress of disease is not linear in each patient during his/her course of disease: some months they have a slow decrease in function and during other periods they have a more rapid progress. There can be a variable arrest of symptoms for months or even years in some patients, but the survival from first symptoms till death is usually 2-5 years, while 10 % have a longer survival. The disease appears in 65% of males, in 35 % of females. The course is often slower in females than in males.

Together, this makes pre-treatment disease progression and disease duration of very limited predictive value in indicating response to treatment of this very heterogeneous disease. Indeed, this small pilot safety study was open to patients of a deliberately heterogeneous background to ensure the relevance of the outcomes to most ALS etiologies. The patients were selected to have different clinical characteristics with respect to the mixture of rate of progress, of age and sex, the type of dysfunction (genetic or sporadic disease, flaccid or spastic paresis, talking or swallowing problems and rate of muscle weakness/muscle atrophy/grade of spasticity).

What is beyond dispute is that, whatever the etiology and clinical characteristics of the patient, once ALS is diagnosed disease progression is NEVER REVERSED so ANY indication of such a response in the outcome data is important. As this small pilot study was designed primarily as a safety and tolerability study for a heterogeneous patient cohort we have, accordingly, been very cautious in attaching significance to the observations of changes in the ALSFRS-R scores despite a positive response from all treated patients.

Notwithstanding the above discussion, as requested we have now included data on the ‘Years since onset of disease symptoms’ for each patient in a new Table 4 on Page 23/24 in the Results section (necessitating renumbering of all Tables). 

2. Effectiveness of treatment should be established by the demonstration of a treatment effect on function in daily activities such as ALFRS-R. However, given the open-label trial, it would have been helpful to include quantitative motor function as an objective measurement.

Response: Quantifiable muscle strength and muscle enzymes depend on the amount of muscle exercise achieved by individuals from day to day, this varies from day to day and patient to patient. Some ALS patients do not even have major peripheral muscle dysfunction but have swallowing or spasticity problems dominating. Most of the challenges in quantitating motor function in ALS patients therefore relate to the disease’s remarkably protean nature. Such heterogeneity adds a challenge that is not present for many other neuromuscular conditions. The difficulties in assessing motor function in ALS patients are described in detail in this review: Rutkove, S.B. Clinical Measures of Disease Progression in Amyotrophic Lateral Sclerosis. Neurotherapeutics 12, 384–393 (2015). https://doi.org/10.1007/s13311-014-0331-9

We have used two of the most reliable and widely used clinical rating scales for ALS in this study: the ALSFRS-R, which is simple to use in patients with muscle and swallowing problems, and the Norris scale, which is more detailed and elucidates a more comprehensive picture of patient functions. Of relevance here, and described on Page 36 Lines 623-625, many of the assessments made as part of ALSFRS-R and Norris (both are Functional Rating Scores) relate to both gross and fine motor skills, e.g., handwriting, speech, swallowing, respiration, dressing, walking, turning in bed, cutting food, climbing stairs, etc., and the scores in these categories show close agreement with objective measures of muscle strength (reference 30).

It should also be noted that this is the first early and small initial safety study of the ILB® in ALS and a more comprehensive testing regime can be incorporated into future larger scale placebo-controlled trials with efficacy as a primary outcome.

3. It is not a scientific expression to state that a patient's vitality has increased, especially in an open-label trial.

Response: Vitality is related to the patient’s perceived levels of energy and fatigue. This was specifically assessed and subjectively quantified as part of the sensory and autonomic symptom panel (tiredness and fatigue subscores). When the scores for these categories were compared before and after treatment, there was a decrease of 25% (P<0.02). This has now been better defined and clarified on Page 35 Lines 600-610.

4. FDA recommends the consideration of add-on designs, in which a treatment previously shown to be effective for the treatment of ALS is given to all patients participating in the trial, with patients randomized to the added investigational drug or added placebo. Too much justification for open-label trial should be avoided. 

Response: We believe that detailing the steps we have taken in the trial design to ensure the generation of meaningful proof of concept data from a small patient cohort are important to those readers who may not be fully cognisant of the challenges in developing drugs for rare orphan diseases with fatal outcomes. However, we accept that we may have laboured this point somewhat. Accordingly, we have shortened this first paragraph in the discussion on Pages 32-33 Lines 541-548 to make it more specific. 

5. In addition, it should be explained in detail that patients taking riluzole are not eligible to participate in this trial.

Response: Motor neuron dysfunction and degeneration in ALS involves several pathogenic mechanisms, which include disturbed energy metabolism, cytoskeletal abnormalities, changes in transcription, glutamate excitotoxicity, glial hyperactivation and reduced glutamate uptake. In our in vitro models we found that the ILB® is able to modulate molecular processes relevant to ALS by modulating the biological effects of growth factors, with multiple downstream consequences. In glial cells the ILB® reduces oxidative stress, mitochondrial dysfunction and glial activation. It also promotes cell survival, differentiation, MBP expression and glutamate uptake. In neuronal cultures ILB® reduces neuronal cell death, oxidative stress and mitochondrial dysfunction. In neurons the compound also promotes differentiation, neurite outgrowth and cellular homeostasis. Most importantly glutamate production and release in neuronal cells is NOT affected by ILB®. Rather, ILB® enhances glutamate uptake by astrocytes. Glutamate production and release (followed by Ca+-mediated signalling cascades) in neuronal cells is one of the most important factors required for synaptic remodelling necessary for regeneration and repair of neuronal networks affected in ALS (and indeed all neurodegenerative diseases). This process is controlled by the growth factors modulated by ILB® and leads to neurite outgrowth and differentiation of these cells. This is the process that will lead to a measurable functional benefit in the nervous system and clinical improvement in the patients with ALS. 

Riluzole and lamotrigine are drugs that affect several neurotransmitter systems and signalling pathways (see references 12,13 and 14 in the manuscript). In vitro studies and patients side effect profiles of the drug demonstrate that riluzole/lamotrigine affects glutamatergic signalling in neurones and most importantly inhibit Ca+-dependent downstream signalling in these cells. This means that riluzole and lamotrigine inhibit some of the same mechanisms that are pivotal for the restoration of neural function induced by ILB®. In summary, based on the established mechanism of action of our drug, it is clear that riluzole/lamotrigine will interfere with key ILB® actions and prevent the beneficial effects in patients.

 As requested, this has now been set out in detail on Pages 8-9 Lines 164-195 of the revised manuscript.

6. Pain and autonomic dysfunction are not typical neurological findings in ALS patients, why did you assess these symptoms?

Response: In fact, 30-50 % of ALS patients have pain and other sensory and autonomic symptoms and dysfunction during the course of their ALS disease. This is well described in the following review and editorial:

Chiò A, Mora G, Lauria G. Pain in amyotrophic lateral sclerosis. Lancet Neurol. 2017 Feb;16(2):144-157. doi: 10.1016/S1474-4422(16)30358-1. Epub 2016 Dec 8. PMID: 27964824.

Vucic S. Sensory and autonomic nervous system dysfunction in amyotrophic lateral sclerosis. Neuropathol Appl Neurobiol. 2017 Feb;43(2):99-101. doi: 10.1111/nan.12336. PMID: 27333192.

The subjective occurrence of pain, sensory and autonomic complaints and symptoms are easily rated as we have done in this study and this monitoring is important to the patients and next of kin. The information is used to inform treatment options for individual patients. To clarify for the reader, we have re-named these symptoms as Subjective Sensory and Autonomic Symptoms on Page 20 Line330. 

Below are some of our publications that describe the impact of these symptoms and our contribution to their investigation:

• Olsson AG, Graneheim Hellgren U, Persson LI, Strand S. Factors that facilitate and hinder the manageability of living with amyotrophic lateral sclerosis in both patients and next of kin. Palliative Support Care 2013;11(3):183-9

• Olsson, AG, Markhede I, Strand S, Persson LI. Well-being in patients with amyotrophic lateral sclerosis and their next of kin over time. Acta Neurol Scand 2010;121:244-250.

• Olsson AG, Markhede I, Strang S, Persson LI. Differences in quality of life modalities give rise to needs of individual support in patients with ALS and their next of kin. Palliative Support Care 2010; 8:75-86.

7. I can't find out the details of the scores for severity of the autonomic and sensory symptoms, but are vertigo and fatigue classified autonomic or sensory symptoms?

Response: Vertigo is often a symptom of an autonomic neuropathy but may also have a sensory causation. Similarly, fatigue is strongly associated with autonomic dysfunction.

The 16 symptom categories related to autonomic/sensory dysfunction that were evaluated in this study were as follows: pain; paresthesia; coldness; sweating; constipation; frequency of faeces; frequency of urination; urination difficulty, numbness; fatigue; tiredness; insomnia; arrhythmia; tachycardia; frequent waking; vertigo/dizziness.

This is now better detailed in the Methods section now entitled Subjective Rating of Sensory and Autonomic Symptoms on Page 20-21, Lines 334-342.

8. To investigate the correlation between the change of serum HGF and clinical measures such as ALSFRS-R would support the hypothesis that increased serum HGF improves motor function of patients with ALS.

Response: This small study has not generated data that can establish a causal relationship between HGF levels and change in ALSFRS-R scores. We note that clinical trials delivering recombinant HGF have achieved serum levels of 20,000 pg/mL with measurable functional effects (e.g., Ido A, Moriuchi A, Numata M, Murayama T, Teramukai S, Marusawa H, Yamaji N, Setoyama H, Kim ID, Chiba T, Higuchi S, Yokode M, Fukushima M, Shimizu A, Tsubouchi H. Safety and pharmacokinetics of recombinant human hepatocyte growth factor (rh-HGF) in patients with fulminant hepatitis: a phase I/II clinical trial, following preclinical studies to ensure safety. J Transl Med. 2011 May 8;9:55. doi: 10.1186/1479-5876-9-55. PMID: 21548996; PMCID: PMC3112439). Our observation that ILB® treatment elevates serum levels to 41,691 pg/mL suggest that the HGF levels achieved in this cohort of patients with ALS are more than sufficient to activate downstream cellular signalling through the C-Met receptor, with beneficial functional effects as indicated. Larger studies with variable doses of ILB® versus placebo lie in the future and these may allow such a link to be made.

Minor issues

9. What subscores of the ALSFRS-R have improved in ALS patients? 

Response: Since this cohort of patients was deliberately chosen to be heterogeneous, their functional insufficiencies were also heterogeneous and varied from individual to individual. Accordingly, when the patient data was pooled we saw improvements in all ALSFRS-R subscores, but the extent of improvement in each category varied between individuals. It is therefore meaningless to analyse and compare individual ALSFRS-R subscores in this small cohort of patients.

10. HGF should be spelled out.

Response: We have done this on Page 5 Line 98-99.

11. The authors need to correct writing errors (FCV→FVC, Line226; forced ventilatory capacity→forced vital capacity, Line 294).

Response: We have corrected this on Pages 18 Line 272-273 and Page 21 Line 344.

Reviewer 1

Major issues:

1. Progression rate of ALSFRS-R and disease duration should be included in the background of patients, as the change in ALSFRS-R is strongly influenced by the disease progression.

Response: See response to Editor Point 1.

2. Effectiveness of treatment should be established by the demonstration of a treatment effect on function in daily activities such as ALFRS-R. However, given the open-label trial, it would have been helpful to include quantitative motor function as an objective measurement.

Response: See response to Editor Point 2.

3. It is not a scientific expression to state that a patient's vitality has increased, especially in an open-label trial.

Response: See response to Editor Point 3.

4. FDA recommends the consideration of add-on designs, in which a treatment previously shown to be effective for the treatment of ALS is given to all patients participating in the trial, with patients randomized to the added investigational drug or added placebo. Too much justification for open-label trial should be avoided. In addition, it should be explained in detail that patients taking riluzole are not eligible to participate in this trial.

Response: See response to Editor Point 4 and 5.

5. Pain and autonomic dysfunction are not typical neurological findings in ALS patients, why did you assess these symptoms?

Response: See response to Editor Point 6.

6. I can't find out the details of the scores for severity of the autonomic and sensory symptoms, but are vertigo and fatigue classified autonomic or sensory symptoms?

Response: See response to Editor Point 7.

7. To investigate the correlation between the change of serum HGF and clinical measures such as ALSFRS-R would support the hypothesis that increased serum HGF improves motor function of patients with ALS.

Response: See response to Editor Point 8.

Minor issues

8. What subscores of the ALSFRS-R have improved in ALS patients?

Response: See response to Editor Point 9.

9. HGF should be spelled out.

Response: See response to Editor Point 10.

10.The authors need to correct writing errors (FCV→FVC, Line226; forced ventilatory capacity→forced vital capacity, Line294).

Response: See response to Editor Point 11.

Reviewer 2

1. This is a manuscript reporting the outcome in an early Phase 2 trial. As it stands there are serious issues in the reporting as the paper does not reflect the protocol. For example, the primary endpoints of the protocol are not mentioned here; nor are the success criteria - what is considered acceptable on these endpoints?

Response: The primary and secondary endpoints of the trial were clearly stated on Page 13 Lines 213-215 of the manuscript as being ‘drug safety and tolerability, secondary endpoints relating to drug efficacy and mechanism of action were also included’. These primary and secondary outcomes were also described in more detail in Section 7 of the supplementary document S1 Trial Protocol, in which Appendix One defines the specific risk assessments made and the treatment stopping criteria as ‘clinical findings which constitute a medical significant risk’. This explicitly states this to be defined as: ‘Serious Adverse Events (SAE) that are assessed as possibly or probably related to study treatment, including all risks categorized as of grade C (as stated in Appendix 1). 

As described in the European Medicines Agency document entitled ICH Topic E 9 Statistical Principles for Clinical Trials: ‘In all clinical trials evaluation of safety and tolerability constitutes an important element. In early phases this evaluation is mostly of an exploratory nature, and is only sensitive to frank expressions of toxicity, whereas in later phases the establishment of the safety and tolerability profile of a drug can be characterised more fully in larger samples of subjects’. Accordingly, this small pilot trial gathered data on all adverse effects off the drug and was specifically looking for drug-related SAE. The absence of any drug-related SAE and the tolerability of the drug were deemed to be the primary criteria for success in this study.

In response to the reviewer, we have now added a sentence to the methods section of manuscript on Page 13 Lines 215-216 that describe what we deemed to be appropriate success criteria for these primary outcomes. We have also added a related comment in the Discussion on Page 33 Lines 564-567.

2. Also the trial was stated to recruit 15 patients. Only 13 were apparently approached according to the flow chart (not a CONSORT diagram as the trial is not randomised). So why was the number to be approached changed? Why was the trial not conducted as stated in the protocol? 

Response: Note that we used the 2016 CONSORT diagram relevant to pilot and feasibility trials in Figure 1 with adaptions as recommended by Lancaster, G.A., Thabane, L. Guidelines for reporting non-randomised pilot and feasibility studies. Pilot Feasibility Stud 5, 114 (2019). https://doi.org/10.1186/s40814-019-0499-1.

The planned number of patients to be included in this safety and tolerability study was 15, however due to delays in recruitment, and the fact that no SAEs or other safety concerns were reported in the first 13 patients, it was decided by the Sponsor, along with the local clinical and ethical agencies, to terminate patient recruitment and conclude the study with the 13. 

The intention to widen this first safety study to a conventional placebo-controlled study was always planned and, as the findings at this first pilot study indicated safety, tolerability and a positive clinical response in all of the first 13 patients tested, it was decided to shorten the study from the planned 15 to 13 patients, to save 5 months of study time, complete the data analysis of the first 13 patients, and use this to more rapidly initiate planning of a larger conventional placebo-controlled study.

 This change in recruitment number plus the reason for it is now described in the Results section on Page 22 Lines 382-87.

3. The methods section does not state the aspiration of the study, and the actual planned number is hidden in the protocol.

Response: We clearly stated the planned primary outcomes (aspirations?) of this trial in the methods on Page 6 Lines 113-114 and again on Page 13 Lines 213-215 in the Methods section. We also mention this in the Abstract (Page 3 Lines 32-34), state a hypothesis in the last paragraph of the Introduction (Page 6 Lines 103-106) and I quote from Page 6 Lines 112-114: ‘The clinical trial was a Phase IIa, single-centre, open label, single-arm proof of concept study of a small number of patients, where the primary endpoint was safety and tolerability of subcutaneously administered ILB®.’ 

As requested, we have now described the planned versus actual number of patients recruited to the trial, plus the reason for the change, on Page 22 Lines 382-387 in the Results section.

4. The patient characteristics are a result and not a method - this section including Table 1 needs to lead the results.

Response: As the reviewer suggests we have moved this information to a new Results section entitled ‘Patient demographics and baseline clinical characteristics with renumbered Tables 3 and 4 on Page 23-24.

5. Please explain why the use of statistical tests here is not in line with the protocol which talks of summary measures only. Please give actual p-values and not just p<0.xx when the p-value exceeds 0.001. What work was done to demonstrate that the data were Normally distributed?

Response: As is well established, the summary measures indicated in the trial protocol provide a quick and simple description of the data and include mean, median, mode, minimum value, maximum value, range, standard deviation, etc. It is appropriate to use this data to perform a paired Student t test or ANOVA to compare the mean of two groups of related samples and to evaluate whether the means of the two sets of data are statistically significantly different from each other. These analyses are described in the Statistical Analysis Plan associated with the clinical trial protocol (see Appendix S1). Exact P values have been given (for example, see Table 8 on Page 31). Normal distribution of data was confirmed by comparing medians and means and by plotting histograms of the frequency distribution of the analysis variable.

6. It is incorrect to only report significant results (e.g., Table 7) - please give effect sizes and CI here and report all tests done - what was the hypothesis here?

Response: The trial protocol and the methods section clearly describes all of the measurements taken on these patients and the related raw data is available on-line to the interested reader who wants more in depth information (link provided in the manuscript on Page 26 Lines 442-443 at https://www.clinicaltrialsregister.eu/ctr-search/trial/2017-005065-47/results and available on request from the authors). To report all data pertaining to non-significant differences within the manuscript would seem superfluous and of limited value to most readers. 

The trial hypothesis is clearly stated on Page 5 Lines 103-106 of the Introduction.

7. Dynamite plots (e.g. Fig 4,5) give less information than a table would - especially given the serial nature of the data. Figure 5C is in particular the wrong figure and again selects only the significant differences. 

Response: We respectfully disagree since the source data is available for detailed evaluation by the interested reader on-line at: https://www.clinicaltrialsregister.eu/ctr-search/trial/2017-005065-47/results

and is also available on request from the authors. We feel that the graphical representation of the data we have made illustrate the outcomes in a visual and easily understandable way, particularly for the non-expert reader.

8 Figure 6 would be clearer as a table.

Response: We respectfully suggest that the graphical representation of the data we have made in Figure 6 better illustrates the changes in clinical characteristics before and after treatment than would a large table of numbers.

Reviewer 3

1. This is a concept-of-proof open, uncontrolled study on a very small and clinically heterogeneous ALS population treated with ILB® and followed up for a short time. The molecule is supposed to be effective mainly by stimulating the production of HGF. The molecule and hypothesized mechanism of action is quite innovative and interesting. In addition, new trials in this population are always welcome given the lack of effective treatments for such a severe and incurable condition. Nevertheless, preclinical data about ILB® in neurological disorders are limited and experiments on ALS animal models are lacking. This limitation must be declared by the authors.

Response: As requested, we have added a short section in the Introduction on Page 5 Lines 93-97 to address this limitation.

2. Safety and tolerability as primary endpoint are well documented and particular attention has been paid to any coagulation disorder potentially related to HGF. In my view, nevertheless, no inference can be drawn about clinical advantages. A clinical impression of "increased vitality, decreased spasticity and increased mobility" and even a transient improvement in functional scales have no relevance in such a small uncontrolled trial. Only robust changes in biological markers can be taken into account in this phase of the experimental study. This strong limitation must be stressed by the authors.

Response: We believe that we have been very cautious in making any claims about the efficacy of ILB®, focussing primarily on safety and tolerability and only indicating potential for efficacy. Indeed, we state in the discussion about efficacy on Page 34 Line 586-590: ‘Although the small size and open-label nature of this study, as well as the lack of rigour of the statistical analysis, necessitates an extremely cautious interpretation, the results of the secondary endpoints (including a limited panel of blood biomarkers that are reported here and elsewhere [39]) suggest the potential for treatment-related change in the progression of ALS.’

 Further evidence of drug efficacy in this same cohort of patients is provided by the serum biomarker results (myoglobin and creatine kinase levels reported here and levels of metabolic biomarkers now reported elsewhere: Lazzarino G, Mangione R, Belli A, Di Pietro V, Nagy Z, Barnes NM, Bruce L, Ropero BM, Persson LI, Manca B, Tavazzi B, Lazzarino G, Logan A. ILB® attenuates oxidative/nitrosative stress and mitochondrial dysfunction in patients with Amyotrophic Lateral Sclerosis. Journal of Personlized Medicine 2021, 11, x. Available from: https://doi.org/10.3390/xxxxx In press).

As requested, we have added this reference in the Discussion section [new Reference 39] to indicate the biomarker data generated in this patient cohort and also emphasised this limitation in the Discussion on Page 34 Lines 586-590.

Reviewer 4

1. ALSFRS-R score improves on average of 4 points after 3 months since the first injection of the IMP (and after 2 months after the last injection). Then, how do the authors explain the worsening of FVC values at day 99? Which are the ALSFRS-R items improving?

Response: The FVC values decline at Day 99 along with a resumption of decline in ALSFRS-R and Norris scores at Day 99. As described, ILB® treatment seems to have slowed, but not reversed, the decline in FVC values, presumably because of the more muted response of the complex neuromuscular control of this function. This attenuation in decline of the FVC values is also reflected in the lack of change in the ALSFRS-R sub-score for Respiratory Insufficiency. Please also see Point 9 of my response to the Editor.

2. What is the reference interval for CK values? why there was no blood CK sampling at day 99? Are creatinine values available?

Response: we apologise as we mistakenly defined the creatine kinase units as U/L in old Table 7 when the assay actually measured plasma creatine kinase as ukat/L. We have corrected this by converting the measured ukat/L values of creatine kinase to U/L in new Table 8 on Page 31 and this reveals the change from 421±337 U/L at Day 1 (Pre-treatment) to 364±299 U/L at Day 36 (Post-treatment). In a healthy Swedish adults, the serum CK level is reported to vary with a number of factors (gender, race and activity), but the normal range is 24-204 U/L (the reference interval) and healthy adult values of CK are reported in Reference 41. Higher amounts of serum CK can indicate muscle damage due to chronic disease or acute muscle injury (see Reference 41).

The protocol indicates that no blood samples were taken for biomarkers at Day 99 in order to minimise interventions on this very sick cohort of patients. Creatine levels were measured in these patients but there was no change in the values recorded over the monitoring period (reported in new Reference 39).

3. Why did the authors evaluate autonomic functions? What exactly is the autonomic and sensory rating scale used in the trial?

Response: See Points 6 and 7 of the response to the Editor.

4. The IMP used in this study seems to reverse the natural decline of ALSFRS-R overtime observed in ALS (without affecting objective measures like NFL or FVC). It is established that NFLs are stable throughout the disease, therefore NFL stabilization shown here was expected and it is not a "positive" result. Furthermore, benefit is prompt. Authors should discuss, somewhere in the discussion, the chance of a placebo effect.

Response: The senior Clinical Investigator in this clinical trial has worked over many years with the biochemist (nowadays a neurologist) Lars Rosengren on studies of CSF biomarkers in ALS and, with others, they have found that the progress of disease in ALS is parallel to CSF and blood levels of Neurofilament Light Chain. Please see References 4 and 5 as examples. In addition to the work of the Clinical Investigator, there are numerous other publications describing the diagnostic and prognostic value of NfL in ALS (for example, see the review by Poesen K, Van Damme P. Diagnostic and Prognostic Performance of Neurofilaments in ALS. Front Neurol. 2019 Jan 18;9:1167. doi: 10.3389/fneur.2018.01167. PMID: 30713520; PMCID: PMC6345692). These publications all describe how the levels of NfL correlate with parameters of disease severity, such as the decline in the ALSFRS-R, and can be used to monitor disease progression. Hence, stabilization of NfL levels may be indicative of a ‘positive’ result.

A placebo group is used in randomised controlled trials to control for the placebo effect and to more accurately determine the efficacy of a new treatment. However, it is also well known that a placebo might also be associated with negative effects (the nocebo effect). Interestingly, a strong nocebo effect has been noted in trials of drugs to treat patients with amyotrophic lateral sclerosis, most recently reported in trials evaluating riluzole and edaravone, drugs which have a partially overlapping mechanism of action to ILB® evaluated here. For unknown reasons, the nocebo effect seems to be stronger in patients with amyotrophic lateral sclerosis than in other neurological disorders (see Rato ML, Duarte GS, Mestre T, de Carvalho M, Ferreira JJ. Strong nocebo effects in amyotrophic lateral sclerosis trials might mask conclusions. Lancet Neurol. 2018 Oct;17(10):842. doi: 10.1016/S1474-4422(18)30310-7. Epub 2018 Sep 18. PMID: 30264722) which could, in theory, mask the beneficial effects of ILB®. As the reviewer suggests, we have included a short section on potential placebo effects in patients with ALS in the Discussion on Pages 34-35 Lines 590-595.

5. Accordingly, why did not the authors evaluate QoL at day 99?

Response: The trial protocol indicates that QoL were not evaluated at Day 99 in order to minimise the research burden on this very sick cohort of patients.

---

## [Decision Letter · Decision Letter 1]

24 Aug 2021

PONE-D-21-04468R1

A phase II open label clinical study of the safety, tolerability and efficacy of ILB® for Amyotrophic Lateral Sclerosis

PLOS ONE

Dear Dr. Logan,

Thank you for submitting your manuscript to PLOS ONE. After careful consideration, we feel that it has merit but does not fully meet PLOS ONE’s publication criteria as it currently stands. Therefore, we invite you to submit a revised version of the manuscript that addresses the points raised during the review process.

We look forward to receiving your revised manuscript.

Kind regards,

Tai-Heng Chen, M.D.

Academic Editor

PLOS ONE

Journal Requirements:

Reviewers' comments:

Reviewer's Responses to Questions

**Comments to the Author**

1. If the authors have adequately addressed your comments raised in a previous round of review and you feel that this manuscript is now acceptable for publication, you may indicate that here to bypass the “Comments to the Author” section, enter your conflict of interest statement in the “Confidential to Editor” section, and submit your "Accept" recommendation.

Reviewer #1: (No Response)

Reviewer #3: All comments have been addressed

2. Is the manuscript technically sound, and do the data support the conclusions?

Reviewer #1: Partly

Reviewer #3: Yes

3. Has the statistical analysis been performed appropriately and rigorously? 

Reviewer #1: No

Reviewer #3: N/A

4. Have the authors made all data underlying the findings in their manuscript fully available?

Reviewer #1: Yes

Reviewer #3: Yes

5. Is the manuscript presented in an intelligible fashion and written in standard English?

Reviewer #1: Yes

Reviewer #3: Yes

6. Review Comments to the Author

Reviewer #1: (No Response)

Reviewer #3: In table 1 please specify diagnosis of ALS as "definite" to uniform it to inclusion criteria in the text.

As a general comment, after authors' description of single symptoms included among "sensory and autonomic" I would rather define them as "non motor" since insomnia is questionable as sensory or autonomic dysfunction

7. PLOS authors have the option to publish the peer review history of their article (what does this mean?). If published, this will include your full peer review and any attached files.

Reviewer #1: No

Reviewer #3: No

---

## [Author Response · Author response to Decision Letter 1]

2 Sep 2021

Ann Logan et al. A phase II open label clinical study of the safety, tolerability and efficacy of ILB® for Amyotrophic Lateral Sclerosis

Response to Editor and Reviewers

We appreciate the further detailed and helpful comments of the reviewers that we have again considered carefully. We believe that the revisions we have made in response to these comments have answered their questions and improved the manuscript significantly.

Editor

Given that ALS is a heterogeneous disease, as the author mentions, I am less certain than the authors that their results indicate a positive response from all the treated patients. 

Response: We have tried to be extremely cautious in our interpretation of the efficacy data throughout. In this revised version of the manuscript, we have included new data regarding the apparent effects of ILB on disease progression by comparing expected versus actual changes in ALSFRS-R scores (new Table 7). We have also emphasized the limitations of this study, the potential for placebo effects and the fact that the results only indicate ‘potential’ as a treatment for ALS (Lines 603-609 and 717-723).

The revision raises some additional issues:

Major issues:

1. As the authors emphasize, ALS is a heterogeneous disease. In fact, several previous studies have shown that the progression rate of ALSFRS-R is useful in predicting disease progression. Therefore, a clear explanation why the authors did not stratify the subjects using the progression rate of ALSFRS-R is necessary.

Response: With the extremely small patient group stratification would not have been possible. However, as suggested we have now used the expected progression rate of the ALSFRS-R scores to expand Table 4 to provide more detail for individual patients on disease characteristics and expectations of disease progression rate during the study period (Table 4 in revised manuscript). Furthermore, we have derived the predicted ALSFRS-R score at key post-treatment intervals for each patient based on their pre-trial disease progression rates as described by Taylor et al., 2016 (new reference number 36). Table 7 now describes the predicted versus actual ALSFRS-R scores for each individual patient at Day 36 and 99, based on their pre-treatment rate of disease progression.

While the added data indicate that ILB treatment does indeed slow the disease progression rate in all patients, our interpretation remains very cautious due to the limitations of this small study. 

2. Unfortunately, there is a lack of confidence in the overall ADL scores, as some subjects have an improvement in Norris score of more than 20 points despite a decrease in ALSFRS-R by more than 5 points in Fig 6. How should I interpret that?

Response: Looking at Figure 6 it can be seen that just one patient (Patient 8) has a significant lack of correlation of the 2 scoring systems. While the Norris and ALSFRS-R scores do show a relatively good correlation, the two scoring systems reflect very different scoring parameters. Therefore, differences in the scores are not unexpected and should not be overinterpreted. Nevertheless, we have now removed Figure 6 from the manuscript and replaced it with new Table 7. As suggested by Reviewer 1 and the Editor, we have derived the predicted ALSFRS-R score at key post-treatment intervals for each patient based on their pre-trial disease progression rates as described by Taylor et al., 2016 (new reference number 36). Table 7 now describes the predicted versus actual ALSFRS-R scores for each individual patient at Day 36 and 99, based on their pre-treatment rate of disease progression. We believe that this new table better describes the changes experienced by each of the trial participants. 

3. If the authors ‘deliberately’ selected patients with different backgrounds as described in the response, it should be sufficiently described in the criteria for patient selection in Methods. A detailed explanation as for how the authors eliminated selection bias is needed.

Response: The inclusion/exclusion criteria used for patient selection are described in detail in Table 1 of the Methods section. The heterogenous patient group we recruited reflects the fact that we ‘deliberately’ did NOT use any enrichment methods to obtain a more homogenous patient population but used the prevalent qualifying cases available in the clinic that carried out the clinical trial. 

4. If the ADL scale is unreliable due to the fact that it is an open-label trial and the results of quantitative motor function test do not exist, it is preferred not to mention too much about the efficacy of the drug.

Response: The functional rating scales (ALSFRS-R and Norris) used in this trial are widely used in clinical trial settings. As mentioned previously, the open label nature of trial may indeed enhance possible placebo effects. However, placebo effects cannot explain the degree of change seen in this cohort of patients (see new reference number 41). Despite this, our interpretations regarding efficacy are very cautious, confined and tempered by the study limitations. 

5. Although dizziness and fatigue can be related to autonomic symptoms, it is not common to include them in the same category due to the wide range of mechanisms. 

Response: We have now recategorized these symptoms as non-motor, autonomic and sensory symptoms (Lines 316 and 321). 

Minor issue:

1. In the table 4, the patient's number is different from the other descriptions.

Response: We have corrected this in Table 4 of the manuscript so that patient number is consistent throughout the manuscript.

Reviewer 3

1. In table 1 please specify diagnosis of ALS as "definite" to uniform it to inclusion criteria in the text.

Response: This has been specified in Table 1 as requested. 

2. As a general comment, after authors' description of single symptoms included among "sensory and autonomic" I would rather define them as "non motor" since insomnia is questionable as sensory or autonomic dysfunction.

Response: This has been redefined as requested in the Methods section on Lines 316 and 321.

NOTE: Please note that we have updated reference number 40.

---

## [Decision Letter · Decision Letter 2]

11 Nov 2021

PONE-D-21-04468R2A phase II open label clinical study of the safety, tolerability and efficacy of ILB® for Amyotrophic Lateral SclerosisPLOS ONE

Dear Dr. Logan,

Thank you for submitting your manuscript to PLOS ONE. After careful consideration, we feel that it has merit but does not fully meet PLOS ONE’s publication criteria as it currently stands. Therefore, we invite you to submit a revised version of the manuscript that addresses the points raised during the review process.

We look forward to receiving your revised manuscript.

Kind regards,

Avanti Dey, PhD

Staff Editor

PLOS ONE  

on behalf of 

Tai-Heng Chen, M.D. 

Academic Editor

Journal Requirements:

Additional Editor Comments (if provided):

Thank you for your patience in relaying this decision to you. Unfortunately some reviewers' comments were removed from the previous decision letter, and which we believe may be necessary in revising your manuscript. We have therefore included those omitted comments below (Reviewer 2), and we kindly request that you address these in your revision.

Reviewers' comments:

Reviewer's Responses to Questions

**Comments to the Author**

1. If the authors have adequately addressed your comments raised in a previous round of review and you feel that this manuscript is now acceptable for publication, you may indicate that here to bypass the “Comments to the Author” section, enter your conflict of interest statement in the “Confidential to Editor” section, and submit your "Accept" recommendation.

Reviewer #1: All comments have been addressed

Reviewer #2: (No Response)

2. Is the manuscript technically sound, and do the data support the conclusions?

Reviewer #1: Partly

Reviewer #2: (No Response)

3. Has the statistical analysis been performed appropriately and rigorously? 

Reviewer #1: Yes

Reviewer #2: (No Response)

4. Have the authors made all data underlying the findings in their manuscript fully available?

Reviewer #1: Yes

Reviewer #2: (No Response)

5. Is the manuscript presented in an intelligible fashion and written in standard English?

Reviewer #1: Yes

Reviewer #2: (No Response)

6. Review Comments to the Author

Reviewer #1: Ann Logan et al.

My concerns have been adequately addressed and the more substantial concerns by the other reviewers have also been met.

**Reviewer #2: Thank you for your response to my previous comments. I'm unclear why the sample size is what it is still. Why 15? Is it an absence of any SAEs or dropouts? With what power is one wanting to rule out a given rate of SAEs? Where is this stated a priori in the protocol? An aspiration is what one is looking for - it is not a measure of outcome - so this still remains missing. There is still on statistical hypothesis. Dynamite plots are well known as conveying less information that in a table not least as spread of data is hidden. Why not give actual data? And why not at the very least give n, mu and sd. This is a commonly accepted minimum, even if one does not accept the consensus of statisticians, presented in https://simplystatistics.org/2019/02/21/dynamite-plots-must-die/ If I find Fig 6 unclear and simply a forest of bars that does not provide the evidence to support the text, other readers will too. The issue is to provide the evidence not aesthetic beauty.**

7. PLOS authors have the option to publish the peer review history of their article (what does this mean?). If published, this will include your full peer review and any attached files.

Reviewer #1: No

Reviewer #2: No

---

## [Author Response · Author response to Decision Letter 2]

19 Dec 2021

Ann Logan et al. A phase II open label clinical study of the safety, tolerability and efficacy of ILB® for Amyotrophic Lateral Sclerosis

Response to Editor and Reviewer #2

Editor

We have appreciated the detailed and helpful comments of the reviewers that we have considered carefully throughout the revision process, but we were surprised to receive a fourth round of comments after such a protracted review period. We understand that it is usually desirable to achieve ‘sign off’ from all reviewers but perceive that this may be unobtainable with Reviewer #2. As further comment, some of the authors are very experienced long-standing researchers including one who is acknowledged as amongst the World’s Most Highly Cited Authors and an Editor-in-Chief of a leading pharmacology journal, yet the precedence of a fourth round of comments is new to all of the authors. Nevertheless, we are happy to hear that we have indeed addressed all of their previous concerns relayed to us in our revised manuscript (version 3) and here we once again try to address the further concerns of Reviewer #2. 

We are dismayed by the misleading comments by Reviewer #2 who seems to disregard (or be unaware of) the current clinical trial design and reporting recommendations accepted by your journal (see details below). Although we believe that it is unnecessary, please let us know if you wish us to include the extra paragraph on methodology set out below (yellow highlighted section). With the offer to insert the extra paragraph if considered necessary, we hope the Editor is now in a position to make a final judgement on our submission.

Reviewer #2

1. Thank you for your response to my previous comments. I'm unclear why the sample size is what it is still. Why 15? Is it an absence of any SAEs or dropouts? 

Response: The protocol planned for 15 patients to be recruited into the clinical trial. This report describes the accumulated data from the first 13 patients recruited into the clinical trial, that was halted early due to the drug’s confirmed safety profile as there were no SAE or dropouts during the study (as clearly described in the clinical trial results and in the paper, see Tables). 

This trial is a phase II exploratory trial (see protocol at https://www.clinicaltrialsregister.eu/ctr-search/trial/2017-005065-47/SE#A). The main aims were to explore the safety of the drug in the target population (phase I data already available in a healthy population) and identify potential for therapeutic efficacy of this new treatment for ALS. The results of this exploratory trial will contribute to the informed design of a phase III trial if/when applicable. 

Presumably being well versed in clinical trial methodology, Reviewer #2 is probably well aware that, due to the lack of information on effect sizes, formal power and sample size calculations for phase II trials are difficult and there is no consensus regarding the best methodology. See: 

Nigel Stallard. Optimal sample sizes for phase II clinical trials and pilot studies. Statist. Med. 2012, 31 1031–1042. DOI: 10.1002/sim.4357 

Steven A. Julious. Sample size of 12 per group rule of thumb for a pilot study. Pharmaceut. Statist. 2005; 4: 287–291 DOI: 10.1002/pst.185.

The issue is even more complex for such exploratory (or pilot studies) when the disease is a rare disease such as ALS. Hence the recommendation of the SPIRIT 2013 statement for clinical trial design:

An-Wen Chan et al., SPIRIT 2013 explanation and elaboration: guidance for protocols of clinical trials. BMJ Research Methods and Reporting. BMJ 2013;346:e7586 doi: 10.1136/bmj.e7586 

This publication accepts that circumstances can dictate that sample size is not derived by statistical calculations. It was this principle that led to the acceptance of the empirically chosen n=15 by the ethics committee who approved the ALS clinical trial described. Additionally, the n=15 satisfies the recommendations of Julious (see reference above) for exploratory trials (minimum recommended n=12; paper attached). The lack of a formal power analysis and sample size calculation is clearly stated i.n our paper (lines 347-348) as recommended by the SPIRIT 2013 statement.

In our submitted paper we endeavour to describe the trial (from design to completion) with honesty and integrity without trying to embellish the findings with post-hoc power analyses of doubtful value. See:

Yiran Zhang et al., Post hoc power analysis: is it an informative and meaningful analysis? Zhang Y, et al. General Psychiatry 2019;32:e100069. doi:10.1136/gpsych-2019-100069 

However, we understand that Reviewer #2 does not want to allow publication of the study without power calculations and sample size estimates. Therefore, we believe it is an editorial decision whether the following paragraph should be included at the end of our statistical analysis: 

“The trial is a phase II exploratory trial. There was no formal sample size estimate or power calculation. The sample size was determined empirically and reflects the exploratory nature of the trial and the rarity of ALS (which is an orphan disease; Chan et al., SPIRIT 2013 statement1). However, at the insistence of the reviewers and editor we have explored whether the trial design would have been admissible using validated approaches to phase II clinical trial design (Jung et al2 and Lin et al3). For design assumptions in this exploration we did NOT use the actual outcome of the trial but relied on what was known in the literature about ALS trial outcomes at the time (high placebo effect and acceptable SAE rate for riluzole) which would have been a reasonable, albeit ambitious, expectation for a new drug. Fleming’s two-stage design (Jung et al2) was used (software made available by the University of North Carolina at Chapel Hill, http://cancer.unc.edu/biostatistics/program/ivanova/FlemingsTwoStageDesign.aspx ) with the following assumptions: the null hypothesis that the response rate is 0.3 (based on the observed placebo effect in ALS trials). The drug will be regarded as good if the desirable effect (positive response = at least no decline in ALSFRS score) is observed in at least 70% of the patients. We expect the type I error rate of 0.05 and 80% power when the positive response rate is 0.7. The results of the analysis are summarised in Table X below. 

Table X. Exploration of trial design and power calculations. Where n is the total number of subjects; n1 is the number of subjects accrued during stage 1; a1, if a1 or fewer responses are observed during stage 1, the trial is stopped early for futility; b1, if b1 or more responses are observed during stage 1, the trial is stopped early and H0 is rejected; a2, if a2 or fewer responses are observed by the end of stage two, then no further investigation of the drug is warranted; b2, b2 = a2 + 1, if b2 or more responses are observed by the end of stage two, H0 is rejected; EN0 is the expected sample size for the trial when response rate is 0.3; EN1 is the expected sample size for the trial when response rate is 0.7. 

n n1 a1 b1 a2 b2 Type 1 Error Power EN0 EN1 Comment

10 7 3 5 5 6 0.0498 0.825 7.2917 7.6807 Minimax

10 6 2 5 5 6 0.0472 0.8379 6.979 8.0374 Minimax

10 7 3 5 5 6 0.0498 0.825 7.2917 7.6807 Minimax

11 5 1 4 6 7 0.0435 0.8136 7.646 7.646 Optimal

13 5 2 5 6 7 0.0408 0.815 6.2852 10.3508 

14 5 2 4 7 8 0.0439 0.8064 6.1907 7.7783 Optimal

14 5 2 4 7 8 0.0439 0.8064 6.1907 7.7783 Optimal

From Table X above it is clear that a trial design including a total of 15 patients is adequate to test the possible efficacy of the drug. It is also clear that stopping the trial early, when 13 patients were recruited did not affect the power of the trial. 

Using the BOP2: Bayesian Optimal Phase II Design software (available from the University of The University of Texas MD Anderson Cancer Center: https://www.trialdesign.org/one-page-shell.html#BOP2 ; Lin et al3) we have checked the validity of our design with one interim analysis allowed. The expectations for efficacy outcome were the same as described above (placebo effect: 30% positive outcome, accepted good drug effect: 70% positive outcome). Additionally, the accepted maximum rate for SAE was 5% (observed with Riluzol (Bensimon et al4) for the null hypothesis. The expectation was that with ILB® we have no SAE (stated stopping criteria 1 patient with SAE possibly related to the drug). We also assume that the efficacy and toxicity of the drug are not related and expect a type I error rate at or below 0.05. With these assumptions and with a maximum of 13 patient enrolled, the results of the 10000 simulations are shown in Table Y below. The power of this trial would be: 0.814. 

 

Table Y. BOP2: Bayesian Optimal Phase II Design with Simple and Complex Endpoints

BOP2 PID: 960; Version: V1.4.10.0; Last Updated: 8/24/2021 (https://www.trialdesign.org/one-page-shell.html#BOP2, used on 07/12/2021). 

Number of patients treated Stop if # response <= OR # toxicity >=

7 2 1

13 6 1

The analysis would indicate that the empirically chosen sample size (with reasonable expectations based on existing knowledge at the time) would have been deemed adequate for a phase II trial. These estimates were produced at the request of reviewers and the editor and are not meant to be a post hoc power analysis based on the outcome of the trial (Zhang et al5).”

References

1An-Wen Chan et al., SPIRIT 2013 explanation and elaboration: guidance for protocols of clinical trials. BMJ Research Methods and Reporting. BMJ 2013;346:e7586 doi: 10.1136/bmj.e7586 

2 Sin-Ho Jung et al., Admissible two-stage designs for phase II cancer clinical trials. Statist. Med. 2004; 23:561–569 (DOI: 10.1002/sim.1600) 

3Ruitao Lin et al., BOIN12: Bayesian Optimal Interval Phase I/II Trial Design for Utility-Based Dose Finding in Immunotherapy and Targeted Therapies. JCO Precis Oncol 4:1393-1402 

4Gilbert Bensimon et al., Riluzole treatment, survival and diagnostic criteria in Parkinson plus disorders: The NNIPPS Study. Brain 2009: 132; 156–171. doi:10.1093/brain/awn291 

5Yiran Zhang et al., Post hoc power analysis: is it an informative and meaningful analysis? Zhang Y, et al. General Psychiatry 2019;32:e100069. doi:10.1136/gpsych-2019-100069 

2. With what power is one wanting to rule out a given rate of SAEs? Where is this stated a priori in the protocol? 

Response: It is stated in the clinical trial protocol and in the paper that no power calculations were performed. For additional detail see above. 

3. An aspiration is what one is looking for - it is not a measure of outcome - so this still remains missing. 

Response: The ‘aspiration’ of the phase II exploratory trial is to ‘explore’, rather than set expectations out of thin air. The reviewer is setting expectations that are fit for a phase III trial and does not seem to be fully aware of the current clinical trial design and reporting recommendations accepted by the journal (see SPIRIT 2013 statement). 

4. There is still on (sic: no?) statistical hypothesis. 

Response: That is correct. There is no statistical hypothesis because there is no need for it in a phase II trial (see above). 

5. Dynamite plots are well known as (sic: for?) conveying less information that (sic: than?) in (sic: is the in necessary?) a table not least as (sic: because the?) spread of data is hidden. Why not give actual data? And why not at the very least give n, mu and sd. This is a commonly accepted minimum, even if one does not accept the consensus of statisticians, presented in https://simplystatistics.org/2019/02/21/dynamite-plots-must-die/ . If I find Fig 6 unclear and simply a forest of bars that does not provide the evidence to support the text, other readers will too. The issue is to provide the evidence not aesthetic beauty.

Response: 

5.1. The comment above is written in poor English and lacks clarity. It certainly falls short of the expectations of a quality peer review. 

5.2. The link (given as a reference regarding the quality of dynamite plots) is from a 2019 twitter post of Prof Rafael Irizarry. The link does not lead to an active page (Error message: “404 File not found, The site configured at this address does not contain the requested file” checked on the 13/12/2021). Perhaps in peer reviews it would be better to use references that the reviewer has actually read – a catchy title from a famous statistician may not necessarily mean “consensus of statisticians”. Journal articles or book chapters may be a better source for scientific guidance to authors than unchecked links lifted from old twitter posts. 

5.3. It is only Figures 4 and 5 that contain cumulative data from the 13 patients. Both graphs give the mean and standard deviation of the group (which is clearly labelled on the graphs) as expected by the reviewer. The n number is 13 as stated several times in the paper. The actual data included in these figures are given in Tables 7 and 8 for each individual patient. So, the “why not give actual data?” and “why not at the very least give n, mu and sd” can only refer to Figure 6. 

5.4. While it is great to hear that Figure 6 was aesthetically pleasing, we must inform the Reviewer and Editor that this figure was removed from the last submitted revised version of the paper and replaced by Table 7 containing the actual numerical data. 

5.5. It may be worth noting that Figure 6 was NOT a dynamite plot anyway. Figure 6 was “Comparison of ALSFRS-R and Norris scores for individual patients at Visit 2 (V2 at Day 1; prior to first ILB® injection) and Visit 7 (V7 at Day 36; 7 days after last ILB® injection)”. So, it was the actual individual scores of patients measured at the two identified time points. There was nothing hidden, the graph contained the actual data from the patients rather than cumulative means. This was made clear in title and legend of the figure. While one would hope that an average reader would read at least the title of the figure before bemoaning the lack of ‘evidence’, the figure was replaced by Table 7 that is ‘less aesthetically pleasing’ but is clear even to the most superficial reader.

---

## [Decision Letter · Decision Letter 3]

21 Jan 2022

PONE-D-21-04468R3A phase II open label clinical study of the safety, tolerability and efficacy of ILB® for Amyotrophic Lateral SclerosisPLOS ONE

Dear Dr. Logan,

Thank you for resubmitting your manuscript to PLOS ONE for further review. We have carefully considered your responses to the reviewers and to the previous editor. The concern of one reviewer about sample size remains. However, thank you for supplying a detailed paragraph defending your position.  I would suggest that the way forward may be for you to include the contents of this paragraph as an additional supplementary file, together with a brief sentence in the main body of the paper highlighting the difference in number recruited from that proposed in the protocol, and your justification for this.  Therefore, we invite you to submit a revised version of the manuscript that addresses these changes.

We look forward to receiving your revised manuscript.

Kind regards,

Antony Bayer

Academic Editor

PLOS ONE

Journal Requirements:

Reviewers' comments:

Reviewer's Responses to Questions

**Comments to the Author**

1. If the authors have adequately addressed your comments raised in a previous round of review and you feel that this manuscript is now acceptable for publication, you may indicate that here to bypass the “Comments to the Author” section, enter your conflict of interest statement in the “Confidential to Editor” section, and submit your "Accept" recommendation.

Reviewer #1: All comments have been addressed

2. Is the manuscript technically sound, and do the data support the conclusions?

Reviewer #1: Partly

3. Has the statistical analysis been performed appropriately and rigorously? 

Reviewer #1: Yes

4. Have the authors made all data underlying the findings in their manuscript fully available?

Reviewer #1: Yes

5. Is the manuscript presented in an intelligible fashion and written in standard English?

Reviewer #1: Yes

6. Review Comments to the Author

Reviewer #1: (No Response)

7. PLOS authors have the option to publish the peer review history of their article (what does this mean?). If published, this will include your full peer review and any attached files.

Reviewer #1: No

---

## [Author Response · Author response to Decision Letter 3]

28 Jan 2022

Response to Editor 

We have appreciated the detailed and helpful comments of the editor and reviewers that we have considered carefully throughout the revision process.

We have carefully evaluated the most recent (fifth round) comments of the Editor and have added a new S2 Appendix and a short section in the Methods (Statistical analysis, P18-19, Lines 345-350) regarding the validity of the patient number evaluated as requested.

---

## [Decision Letter · Decision Letter 4]

21 Feb 2022

PONE-D-21-04468R4A phase II open label clinical study of the safety, tolerability and efficacy of ILB® for Amyotrophic Lateral SclerosisPLOS ONE

Dear Dr. Logan,

Thank you for submitting the latest version your manuscript to PLOS ONE and for including the new information about sample size. As you will see from the statistical referee's comments below, he is asking about the justification for the choice of 70% for patients showing a positive response. When and why was this cutoff chosen? Therefore, we invite you to submit a revised version of the manuscript that addresses this point.

We look forward to receiving your revised manuscript.

Kind regards,

Antony Bayer

Academic Editor

PLOS ONE

Journal Requirements:

Reviewers' comments:

Reviewer's Responses to Questions

**Comments to the Author**

1. If the authors have adequately addressed your comments raised in a previous round of review and you feel that this manuscript is now acceptable for publication, you may indicate that here to bypass the “Comments to the Author” section, enter your conflict of interest statement in the “Confidential to Editor” section, and submit your "Accept" recommendation.

Reviewer #2: (No Response)

2. Is the manuscript technically sound, and do the data support the conclusions?

Reviewer #2: (No Response)

3. Has the statistical analysis been performed appropriately and rigorously? 

Reviewer #2: (No Response)

4. Have the authors made all data underlying the findings in their manuscript fully available?

Reviewer #2: (No Response)

5. Is the manuscript presented in an intelligible fashion and written in standard English?

Reviewer #2: (No Response)

6. Review Comments to the Author

Reviewer #2: I am curious about supplement 2 - where did the numbers for efficacy etc come from - were they decided a priori? I can find no mention of 70% in the protocol - please can you point me to the a priori decsions on minimum acceptable efficacy and maximal acceptable toxicity

7. PLOS authors have the option to publish the peer review history of their article (what does this mean?). If published, this will include your full peer review and any attached files.

Reviewer #2: No

---

## [Author Response · Author response to Decision Letter 4]

25 Feb 2022

Response to the Editor

We have made 4 changes to the manuscript (highlighted in red in the tracked manuscript. 

1. On Page 1 Line 7-9: At his request we have removed Bernardo Ropero from the author list. 

2. On Page 2 Line 27 we have corrected the email address of the corresponding author.

3. On Page 18 Line 348 we have corrected the supplementary appendix notation.

4. On Page 36 Line 675 we have removed reference to the funding for the study.

5. On Page 46 Lines 915-917 we have corrected the list of supplementary material.

Response to Reviewer #2

Reviewer #2: I am curious about supplement 2 - where did the numbers for efficacy etc come from - were they decided a priori? I can find no mention of 70% in the protocol - please can you point me to the a priori decisions on minimum acceptable efficacy and maximal acceptable toxicity

We have carefully evaluated the most recent comments of the Reviewer #2 and have added a new section in the S2 Appendix (Lines 18-42, Lines 74-76 and 4 new references on Lines 104-110) regarding the chosen response rates and toxicity as requested.

---

## [Editor Report · Decision Letter 5]

5 Apr 2022

A phase II open label clinical study of the safety, tolerability and efficacy of ILB® for Amyotrophic Lateral Sclerosis

PONE-D-21-04468R5

Dear Dr. Logan,

Thank you for your your further revised manuscript and for your response to the reviewer’s final concerns. We are pleased to inform you that your manuscript has been judged scientifically suitable for publication and will be formally accepted for publication once it meets all outstanding technical requirements.

Kind regards,

Antony Bayer

Academic Editor

PLOS ONE
---

## [Editor Report · Acceptance letter]

28 Apr 2022

PONE-D-21-04468R5 

A phase II open label clinical study of the safety, tolerability and efficacy of ILB for Amyotrophic Lateral Sclerosis 

Dear Dr. Logan:

I'm pleased to inform you that your manuscript has been deemed suitable for publication in PLOS ONE. Congratulations! Your manuscript is now with our production department. 

Kind regards, 

on behalf of

Professor Antony Bayer 

Academic Editor

PLOS ONE